# Genetic requirement of *dact1/2* to regulate noncanonical Wnt signaling and *calpain 8* during embryonic convergent extension and craniofacial morphogenesis

Shannon H Carroll[1,2,3], Sogand Schafer[1,2], Kenta Kawasaki[2,3], Casey Tsimbal[2,3], Amelie M Jule[4], Shawn A Hallett[2,3], Edward Li[2], Eric C Liao[1,2,3]*

[1]Center for Craniofacial Innovation, Children's Hospital of Philadelphia Research, Institute, Children's Hospital of Philadelphia, Philadelphia, United States; [2]Division of Plastic and Reconstructive Surgery, Department of Surgery, Children's Hospital of Philadelphia, Philadelphia, United States; [3]Shriners Hospital for Children, Tampa, United States; [4]Department of Biostatistics, Harvard T.H. Chan School of Public Health, Boston, United States

**\*For correspondence:**
liaoce@chop.edu

**Competing interest:** The authors declare that no competing interests exist.

## eLife Assessment

This in several parts **valuable** study confirms the roles of Dact1 and Dact2, two factors involved in Wnt signaling, during zebrafish gastrulation and demonstrates their genetic interactions with other Wnt components to modulate craniofacial morphologies. Unfortunately, there are several limitations associated with the study, making it challenging to distinguish the primary and secondary effects of each factor, and their roles in craniofacial morphogenesis. The findings of a new potential target of dact1/2-mediated Wnt signaling are potentially of value; however, experimental evidence supporting their functional significance remains **incomplete** due to inconsistent results and limitations inherent to the overexpression approach.

**Abstract** Wnt signaling plays crucial roles in embryonic patterning including the regulation of convergent extension (CE) during gastrulation, the establishment of the dorsal axis, and later, craniofacial morphogenesis. Further, Wnt signaling is a crucial regulator of craniofacial morphogenesis. The adapter proteins Dact1 and Dact2 modulate the Wnt signaling pathway through binding to Disheveled. However, the distinct relative functions of Dact1 and Dact2 during embryogenesis remain unclear. We found that *dact1* and *dact2* genes have dynamic spatiotemporal expression domains that are reciprocal to one another suggesting distinct functions during zebrafish embryogenesis. Both *dact1* and *dact2* contribute to axis extension, with compound mutants exhibiting a similar CE defect and craniofacial phenotype to the *wnt11f2* mutant. Utilizing single-cell RNAseq and an established noncanonical Wnt pathway mutant with a shortened axis (*gpc4*), we identified *dact1/2*-specific roles during early development. Comparative whole transcriptome analysis between wildtype and *gpc4* and wildtype and *dact1/2* compound mutants revealed a novel role for *dact1/2* in regulating the mRNA expression of the classical calpain *capn8*. Overexpression of *capn8* phenocopies *dact1/2* craniofacial dysmorphology. These results identify a previously unappreciated role of *capn8* and calcium-dependent proteolysis during embryogenesis. Taken together, our findings highlight the distinct and overlapping roles of *dact1* and *dact2* in embryonic craniofacial development, providing new insights into the multifaceted regulation of Wnt signaling.

## Introduction

Wnt signaling is a crucial regulator of embryogenesis through its regulation of body axis patterning, cell fate determination, cell migration, and cell proliferation (*Logan and Nusse, 2004*; *Steinhart and Angers, 2018*; *Mehta et al., 2021*). Current mechanistic understanding of Wnt signaling during embryogenesis includes an extensive catalog of ligands, receptors, co-receptors, adaptors, and effector molecules (*Clevers and Nusse, 2012*; *Niehrs, 2012*; *Loh et al., 2016*; *Mehta et al., 2021*). The intricate spatiotemporal integration of Wnt signaling combinations is an important focus of developmental biology and tissue morphogenesis (*Petersen and Reddien, 2009*; *Clevers and Nusse, 2012*; *Loh et al., 2016*; *Wiese et al., 2018*). Disruptions of Wnt signaling-associated genes lead to several congenital malformations which often affect multiple organ systems given their pleotropic developmental functions (*Hashimoto et al., 2014*; *Shi, 2022*). Craniofacial anomalies are among the most common structural congenital malformations and genes in the Wnt signaling pathway are frequently implicated (*Ji et al., 2019*; *Reynolds et al., 2019*; *Huybrechts et al., 2020*).

Genetic approaches in zebrafish have identified a number of key Wnt regulators of early development, with gastrulation and craniofacial phenotypes (*Brand et al., 1996*; *Hammerschmidt et al., 1996*; *Heisenberg et al., 1996*; *Piotrowski et al., 1996*; *Solnica-Krezel et al., 1996*; *Schilling and Le Pabic, 2009*). The *silberblick* (*slb*) mutant, later identified as a *wnt11f2* mutant allele, exhibits gastrulation and midline craniofacial phenotypes that encompassed aspects of multiple mutant classes. During early segmentation in the somite stage, the *wnt11f2* mutant developed a shortened anterior–posterior axis and partially fused eyes (*Heisenberg et al., 1996*). Subsequently, as the cranial prominences converge and the ethmoid plate (EP) formed, instead of a fan-shaped structure observed in wildtype embryos, the *wnt11f2* mutant formed a rod-like EP with a significant deficiency of the medio-lateral dimension (*Heisenberg et al., 1996*; *Heisenberg and Nüsslein-Volhard, 1997*). Another mutant kny*pek* (*kny*), identified as having a nonsense mutation in *gpc4*, an extracellular Wnt co-receptor, was identified as a gastrulation mutant that also exhibited a shortened body axis due to a defect in embryonic convergent extension (CE) (*Solnica-Krezel et al., 1996*). In contrast to the *slb/wnt11f2* mutant, the *gpc4* mutant formed an EP that is wider in the medio-lateral dimension than the wildtype, in the opposite end of the EP phenotypic spectrum compared to *wnt11f2* (*Topczewski et al., 2001*; *Rochard et al., 2016*). These observations beg the question of how defects in early patterning and CE of the embryo may be associated with later craniofacial morphogenesis. The observation that *wnt11f2* and *gpc4* mutant share similar CE dysfunction and axis extension phenotypes but contrasting craniofacial morphologies (*Heisenberg and Nüsslein-Volhard, 1997*) supports a hypothesis that CE mechanisms regulated by these Wnt pathway genes are specific to the temporal and spatial contexts during embryogenesis.

Dact (aka Frodo, Dapper) are scaffolding proteins that regulate Dishevelled (Dvl)-mediated Wnt signaling, both positively and negatively (*Cheyette et al., 2002*; *Gloy et al., 2002*; *Waxman et al., 2004*; *Gao et al., 2008*; *Wen et al., 2010*; *Ma et al., 2015*; *Lee et al., 2018*). Dact proteins bind directly to Dvl (*Gloy et al., 2002*; *Brott and Sokol, 2005*; *Lee et al., 2018*) and interact with and inhibit members of transforming growth factor beta (TGF-β) and Nodal signaling pathways (*Zhang et al., 2004*; *Su et al., 2007*; *Meng et al., 2008*; *Kivimäe et al., 2011*). In chick and *Xenopus*, *Dact2* and *Dact1* (respectively) are expressed in the neural folds during neural crest delamination and are important in epithelial–mesenchymal transition (EMT), Wnt signaling, and TGF-β signaling (*Hikasa and Sokol, 2004*; *Schubert et al., 2014*; *Rabadán et al., 2016*). In mouse embryos, Dact1 is expressed predominantly in mesodermal tissues, as well as ectodermal-derived tissues (*Hunter et al., 2006*), and ablation of Dact1 results in defective EMT and primitive streak morphogenesis, with subsequent posterior defects (*Wen et al., 2010*). Mouse embryonic Dact2 expression has been described in the oral epithelium and ablation of *Dact2* causes increased cell proliferation (*Li et al., 2013*) and re-epithelialization in mice (*Meng et al., 2008*) and zebrafish (*Kim et al., 2020*).

Previous experiments using morpholinos to disrupt *dact1* and *dact2* in zebrafish found *dact1* morphants to be slightly smaller and to develop a normal body. In contrast, *dact2* morphants were found to phenocopy described zebrafish gastrulation mutants, with impaired CE, shortened body axis, and medially displaced eyes. Importantly, prior work using morpholino-mediated gene disruption of *dact1* and *dact2* did not examine craniofacial morphogenesis except to analyze *dact1* and *dact2* morphants head and eye shapes under light microscopy (*Waxman et al., 2004*). These experiments were carried out at a time when morpholino was the accessible tool of gene disruption (*Nasevicius*

and Ekker, 2000; Corey and Abrams, 2001; Heasman, 2002). Since CRISPR/Cas9 targeted gene mutagenesis became popularized, many reports of germline mutant phenotypes being discrepant from prior morpholino studies warranted revisiting many of the prior work (Kok et al., 2015) and careful interpretation given the caveats of each technology (Morcos et al., 2015; Rossi et al., 2015). More recently, a zebrafish CRISPR/Cas9 genetic dact2 mutant has been generated and studied, but unlike in the dact2 morphant, no developmental phenotypes were described (Kim et al., 2020).

Here, we investigated the genetic requirement of dact1 and dact2 during embryogenesis and craniofacial development using germline mutant alleles. We found an early developmental role for dact1 and dact2 during gastrulation and body axis elongation. We also characterized the abnormal craniofacial development of the dact1/2 compound mutants. We identified distinct transcriptomic profiles of wildtype, dact1/2, and gpc4 mutants during early development, including finding calpain 8 (capn8) calcium-dependent protease to be ectopically expressed in the dact1/2 mutants. These results elaborate on the cellular roles of dact1/2 and identify capn8 as a novel regulatory candidate of embryogenesis.

## Results

### dact1 and dact2 have distinct expression patterns throughout embryogenesis

To determine the spatiotemporal gene expression of dact1 and dact2 during embryogenesis we performed wholemount RNA in situ hybridization (ISH) across key time points (Figure 1). Given that the described craniofacial phenotypes of the dact2 morphant and the wnt11f2 mutant are similar (Heisenberg and Nüsslein-Volhard, 1997; Waxman et al., 2004), we also performed wnt11f2 ISH to compare to dact1 and dact2 expression patterns.

During gastrulation at 8 hours post-fertilization (hpf; 75% epiboly), some regions of dact1 and dact2 gene expression were shared and some areas are distinct to each dact gene (Figure 1A). Further, dact gene expression was distinct from wnt11f2 in that wnt11f2 expression was not detected in the presumptive dorsal mesoderm. Transcripts of dact1, dact2, and wnt11f2 were all detected in the blastoderm margin, as previously described (Makita et al., 1998; Heisenberg et al., 2000; Gillhouse et al., 2004). Transcripts of dact2, and to a lesser extent dact1, were also detected in the prechordal plate and chordamesoderm (Figure 1A). Additionally, dact2 gene expression was concentrated in the shield and presumptive organizer or Nieuwkoop center along with wnt11f2. This finding is consistent with previously described expression patterns in zebrafish and supports a role for dact1 and dact2 in mesoderm induction and dact2 in embryo dorsalization (Thisse et al., 2001; Gillhouse et al., 2004; Muyskens and Kimmel, 2007; Oteiza et al., 2010). At the end of gastrulation and during somitogenesis the differences in the domains of dact1 and dact2 gene expressions became more distinct (Figure 1B, C). At tailbud stage, dact1 transcripts were detected in the neuroectoderm and the posterior paraxial mesoderm, whereas dact2 transcripts were detected in the anterior neural plate, notochord, and tailbud. Anterior notochord and tailbud expression overlapped with wnt11f2 gene expression (Makita et al., 1998; Heisenberg et al., 2000). The expression of dact2 was unique in that its expression demarcated the anterior border of the neural plate. As dact2 morphants exhibited a craniofacial defect with medially displaced eyes and midfacial hypoplasia (Waxman et al., 2004), we examined dact1 and dact2 expression in the orofacial tissues. At 24 hpf we found some overlap but predominantly distinct expression patterns of dact1 and dact2 with dact1 being more highly expressed in the pharyngeal arches and dact2 being expressed in the midbrain/hindbrain boundary. Both dact1 and dact2 appeared to be expressed in the developing oral cavity. At 48 hpf dact1 expression is consistent with expression in the developing craniofacial cartilage elements, while dact2 expression appears within the developing mouth. The distinct cellular expression profiles of dact1 and dact2 were more clear in histological sections through the craniofacial region at 72 hpf. Utilizing RNAscope ISH, we found that dact2 and the epithelial gene irf6 were co-expressed in the surface and oral epithelium that surround the cartilaginous structures (Figure 1F). This is in contrast to dact1 which was expressed in the developing cartilage of the anterior neurocranium (ANC)/EP and palatoquadrate of the zebrafish larvae (Figure 1F).

We examined the overall expression patterns of dact1, dact2, gpc4, and wnt11f2 using Daniocell single-cell sequencing data (Farrell et al., 2018). In general, we found dact1 spatiotemporal gene

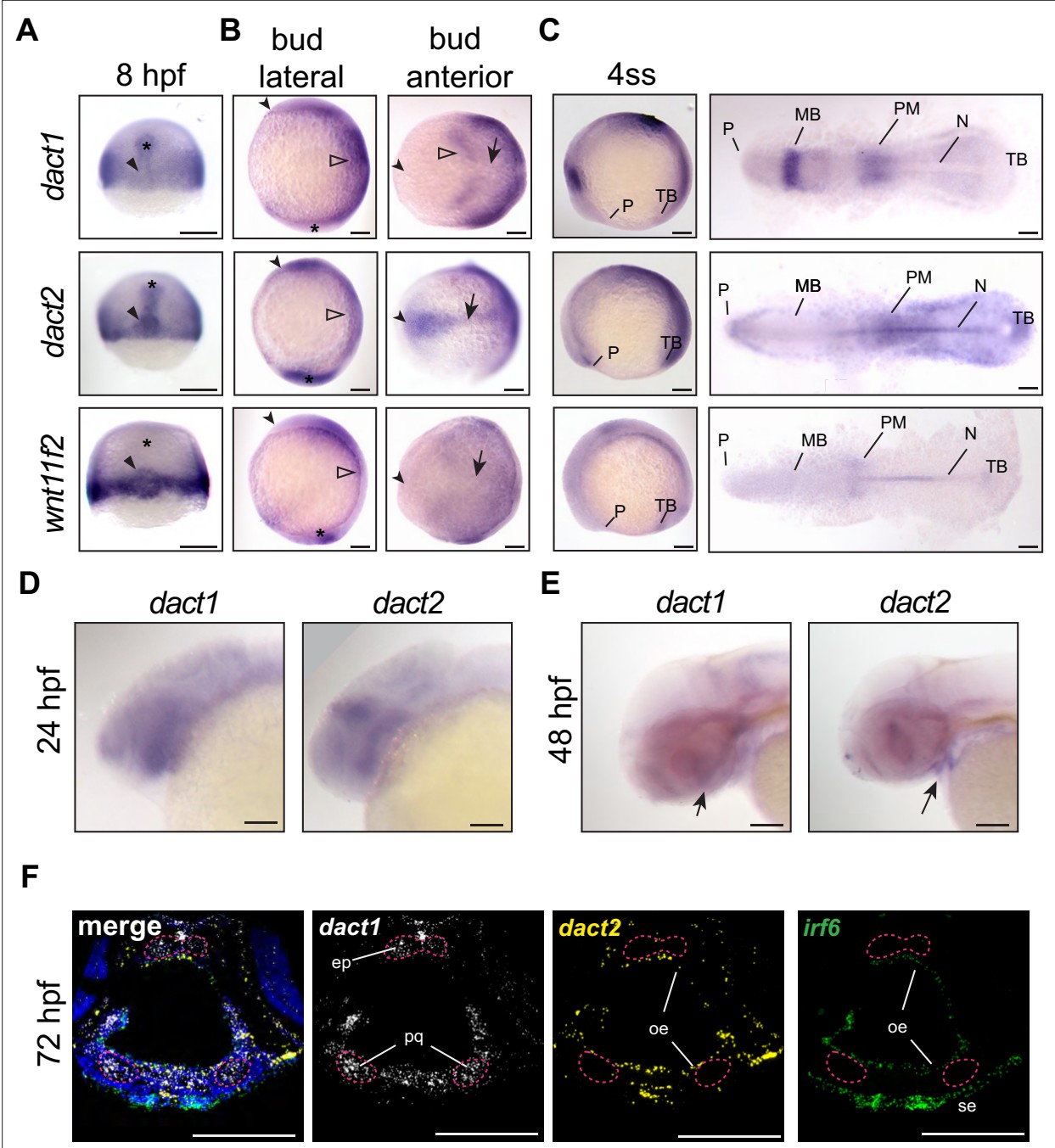

**Figure 1.** Unique and shared *dact1* and *dact2* gene expression domains during zebrafish development. (**A–C**) Representative images of wholemount in situ hybridization showing *dact1*, *dact2*, and *wnt11f2* gene expression patterns. (**A**) At 8 hpf, *dact2* and *wnt11f2* are highly expressed in the dorsal margin and presumptive Nieuwkoop center of the gastrulating embryo, with *dact1* being weakly detected (arrowhead). In contrast to *wnt11f2*, *dact1*, and *dact2* are expressed in the presumptive dorsal mesoderm (asterisk). (**B**) Lateral (anterior to the left of page) and anterior (dorsal side toward top of page) views of bud-stage embryos. *dact2* and *wnt11f2* transcripts are both detected in the tailbud (asterisk) while *dact2* is additionally expressed in the axial mesoderm (arrow). *dact1* gene expression is concentrated to the paraxial mesoderm and the neuroectoderm (open arrowheads). (**C**) Lateral and flat-mount views of 4 ss embryos. *dact2* is expressed in the anterior neural plate and polster (P), notochord (N), paraxial and presomitic mesoderm (PM) and tailbud (TB). In contrast, *dact1* is expressed in the midbrain (MB) and the paraxial and presomitic mesoderm. (**D, E**) Representative lateral (anterior to left of page) images of wholemount in situ hybridization showing *dact1* and *dact2* expression patterns. (**D**) At 24 hpf expression is detected in the developing head. (**E**) At 48 hpf expression is detected in the developing craniofacial structures (arrow). (**F**) Representative images of RNAscope in situ hybridization analysis of *dact1* (white) and *dact2* (yellow) and *irf6* (green) expression in transverse section of 72 hpf embryos. *dact1* is expressed

*Figure 1 continued on next page*

*Figure 1 continued*

in the ethmoid plate (ep) and palatoquadrate (pq) orofacial cartilage, while *dact2* is expressed in the oral epithelium (oe). The epithelial marker *irf6* is expressed in the oe and surface epithelium (se). Dapi (blue). Scale bar: 100 µm.

The online version of this article includes the following figure supplement(s) for figure 1:

**Figure supplement 1.** Daniocell single-cell RNAseq analysis with a display of *dact1*, *dact2*, *gpc4*, and *wnt11f2* in all cell clusters from 3 to 120 hpf of development (https://daniocell.nichd.nih.gov; *Farrell et al., 2018*).

expression to be more similar to *gpc4* while *dact2* gene expression was more similar to *wnt11f2* (*Figure 1—figure supplement 1*). These results of shared but also distinct domains of spatiotemporal gene expression of *dact1* and *dact2* suggest that the *dact* paralogs may have some overlapping developmental functions while other roles are paralog-specific. *dact1* and *dact2* contribute to axis extension and *dact1/2* compound mutants exhibit a CE defect *dact1* and *dact2* are known to interact with *disheveled* and regulate noncanonical Wnt signaling (*Gloy et al., 2002*; *Waxman et al., 2004*; *Gao et al., 2008*; *Wen et al., 2010*; *Ma et al., 2015*; *Lee et al., 2018*) and we have previously described the craniofacial anomalies of several zebrafish Wnt mutants (*Dougherty et al., 2012*; *Kamel et al., 2013*; *Rochard et al., 2016*; *Ling et al., 2017*; *Alhazmi et al., 2021*). Previous work investigated the effect of *dact1* and *dact2* disruption during zebrafish embryogenesis using morpholinos and reported morphant phenotypes in embryonic axis extension and eye fusion (*Waxman et al., 2004*). However, the limitations associated with morpholino-induced gene disruption (*Kok et al., 2015*; *Morcos et al., 2015*; *Rossi et al., 2015*) and the fact that craniofacial morphogenesis was not detailed for the *dact1* and *dact2* morphants, warranted the generation of mutant germline alleles (*Figure 2—figure supplement 1*). We created a *dact1* mutant allele (22 bp deletion, hereafter *dact1−/−*) and a *dact2* mutant allele (7 bp deletion, hereafter *dact2−/−*), both resulting in a premature stop codon and presumed protein truncation (*Figure 2—figure supplement 1A, B*). Gene expression of *dact1* and *dact2* was measured in pooled *dact1−/−*, *dact2−/−*, and *dact1/2−/−* embryos (*Figure 2—figure supplement 1C*). We found a decrease in *dact1* mRNA and an increase in *dact2* mRNA levels in the respective CRISPR single mutants. We hypothesize that *dact2* mRNA levels are maintained or elevated in the *dact2−/−* mutant due to the relative 3′ position of the deletion. In the *dact2−/−* embryos we found a slight increase in *dact1* mRNA levels, suggesting a possible compensatory effect of dact2 disruption. The specificity of the gene disruption was demonstrated by phenotypic rescue of the rod-like EP with the injection of *dact1* or *dact2* mRNA. Injection of *dact1* mRNA or *dact2* mRNA or in combination decreased the percentage of rod-like EP phenotype from near the expected 25% (35% actual) to 2–7% (*Figure 2—figure supplement 1D and E*).

Analysis of compound *dact1* and *dact2* heterozygote and homozygote alleles during late gastrulation and early segmentation time points identified embryonic axis extension anomalies (*Figure 2A, B*). *dact1−/−* or *dact2−/−* homozygotes develop to be phenotypically normal and viable. However, at 12 hpf, *dact2−/−* single mutants have a significantly shorter body axis relative to wildtype. In contrast, body length shortening phenotype was not observed in *dact1−/−* homozygotes. Compound heterozygotes of *dact1+/-; dact2+/-* also developed normally but exhibited shorter body axis relative to wildtype. The most significant axis shortening occurred in *dact1−/−; dact2−/−* double homozygotes with a less severe truncation phenotype in the compound heterozygote *dact1+/-; dact2−/−* (*Figure 2A, B*). Interestingly, these changes in body axis extension do not preclude the compound heterozygous larvae from reaching adulthood, except in the *dact1−/−; dact2−/−* double homozygotes which did not survive from larval to juvenile stages.

Body axis truncation has been attributed to impaired CE during gastrulation (*Tada and Heisenberg, 2012*). To delineate CE hallmarks in the *dact1−/−;dact2−/−* mutants, we performed wholemount RNA ISH detecting genes that are expressed in key domains along the body axis. At bud stage, *dact1−/−; dact2−/−* embryos demonstrate bifurcated expression of *pax2a* and decreased anterior extension of *gsc* expression, suggesting impaired midline convergence and anterior extension of the mesoderm (*Figure 2C*). At the 1–2 somite stage, *zic1*, *pax2a*, and *tbx6* are expressed in neural plate, prospective midbrain and the tailbud, respectively, in both the wildtype and *dact1−/−; dact2−/−* embryos. However, the spacing of these genes clearly revealed the shortening of the antero-posterior body axis in the *dact1−/−; dact2−/−* embryos. Midline convergence is decreased and the anterior border of the neural plate (marked by *zic1* expression) was narrower in the *dact1−/−; dact2−/−* embryos (*Figure 2D*). At the 10-somite stage (ss), *dact1−/−; dact2−/−* embryos

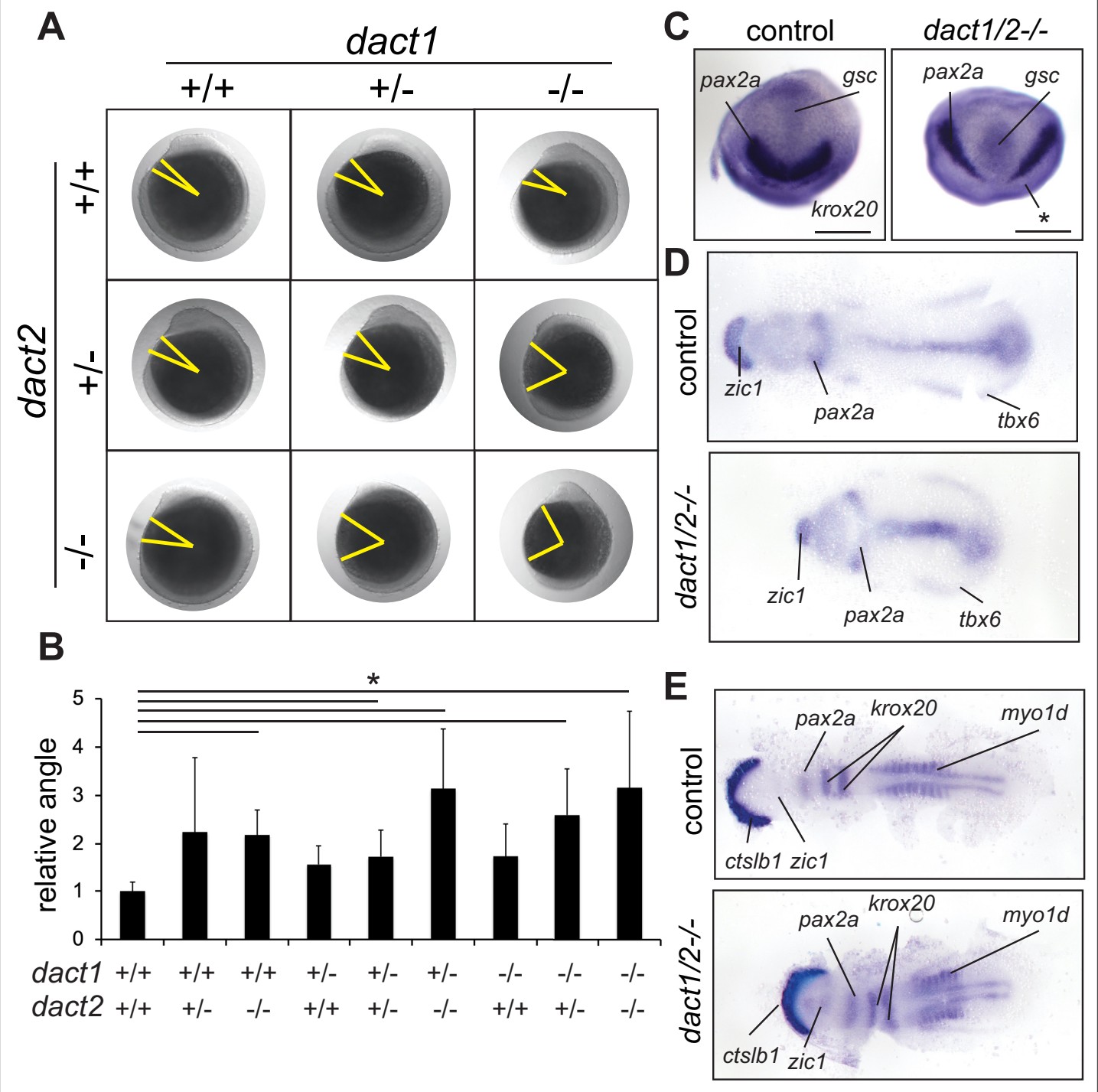

**Figure 2.** Impaired convergent extension in *dact1/2* compound mutants. (**A**) Inter-cross of compound heterozygotes yield embryos with different degrees of axis extension that correspond to the *dact1* and *dact2* genotypes. Representative lateral images of embryos at 12 hpf. The yellow line indicates body axis angle measured from the anterior point of the head, the center of the yolk, to the end of the tail. (**B**) Quantification of body axis angle. Numbers represent the difference in angle relative to the average wildtype embryo. Asterisk indicates genotypes with angles significantly different from wildtype. ANOVA p < 0.5 *n* = 3–21 embryos. Error bars: ± SEM. (**C**) Representative bud stage wildtype and *dact1/2−/−* mutant embryos stained for *gsc* (prechordal plate), *pax2a* (midbrain/hindbrain boundary), and *krox20* (rhombomere 3). Asterisk indicates lack of *krox20* expression in *dact1/2−/−* mutant. Scale bar = 200 μm (**D**) Representative flat mounts of 1–2 ss wildtype and *dact1/2* mutant embryos stained for *zic1* (telencephalon), *pax2a* and *tbx6* (ventrolateral mesoderm). (**E**) Representative flat mounts of 10 ss wildtype and *dact1/2−/−* mutant embryos stained for *ctsl1b* (hatching gland), *zic1*, *pax2a*, *krox20*, and *myo1d* (somites).

*Figure 2 continued on next page*

*Figure 2 continued*

The online version of this article includes the following figure supplement(s) for figure 2:

**Figure supplement 1.** Characterization of CRISPR/Cas9 generated *dact1*−/− and *dact2*−/− mutants.

**Figure supplement 2.** Loss of *dact1* and *dact2* tends to decrease total body length.

demonstrated decreased spacing between *ctslb1* and *pax2a* gene expression, suggesting impaired lengthening of the anterior portion of the embryo. Detection of muscle marker *myo1d* in the *dact1*−/−; *dact2*−/− embryos delineated impaired posterior lengthening as well as reduced somitogenesis, evidenced by the decreased number of somites (*Figure 2E*). These data point to impaired CE of the mesoderm in *dact1/2*−/− double mutants, which resulted in a shorter body axis. The aberrant CE and axis extension in the *dact1/2*−/− phenotypes were similar to findings in other Wnt mutants, such as *slb* and *kyn* (*Heisenberg et al., 2000*; *Topczewski et al., 2001*) in that the body axis is truncated upon segmentation.

## *dact1/dact2* compound mutants exhibit axis shortening and craniofacial dysmorphology

Given the defective converge phenotype and shortened axis in the *dact* mutants during gastrulation, we examined the fish at 4 dpf for axis defects and for evidence of defective morphogenesis in the craniofacial cartilages. Craniofacial morphology is an excellent model for studying CE morphogenesis as many craniofacial cartilage elements develop through this cellular mechanism (*Kamel et al., 2013*; *Mork and Crump, 2015*; *Sisson et al., 2015*; *Rochard et al., 2016*). No craniofacial phenotype was observed in *dact1* or *dact2* single mutants (data not shown). However, in-crossing to generate *dact1/2*−/− compound homozygotes resulted in dramatic craniofacial malformation (*Figure 3*). Specificity of this phenotype to *dact1/2* was confirmed via rescue with *dact1* or *dact2* mRNA injection (*Figure 2—figure supplement 1D, E*). The *dact1/2*−/− mutant embryos exhibited fully penetrant midfacial hypoplasia (*Figure 3A*); however, the degree of eye field convergence in the midline varied between individuals. The forebrain protruded dorsally and the mouth opening and ventral cartilage structures were displaced ventrally (*Figure 3A*). Alcian blue staining of cartilage elements revealed severe narrowing of the EP into a rod-like structure in 100% of double mutants, while the ventral cartilage elements were largely unaffected (*Figure 3B*). Notably the trabeculae extending posteriorly from the EP and the rest of the posterior neurocranium exhibit wildtype morphology in *dact1/2*. This *dact1/2*−/− double mutant phenotype is highly similar to that described for *wnt11f2* (*slb*) mutants, a key regulator of noncanonical Wnt signaling and CE (*Kimmel et al., 2001*).

As axis lengthening was found to be affected by loss of *dact1* and *dact2* (*Figure 2A*) we measured body length in 5 dpf *dact1/2* compound mutants. Using the length of the vertebral spine as a measure of body length we found a trend (p = 0.06) toward an effect of *dact1* and *dact2* on shortening of the body length. Similar to axis length during gastrulation/segmentation, the shortening was most pronounced in *dact1/2*−/− double homozygous mutants versus wildtype clutch-mates (*Figure 3C*, *Figure 2—figure supplement 2A, B*).

As *wnt11f2* signals via disheveled and since dact proteins are known to interact with disheveled (*Wong et al., 2003*; *Zhang et al., 2006*; *Kivimäe et al., 2011*), it is suspected that dact has a role in wnt11f2 signaling. Combinatorial gene disruption with morpholinos showed that *dact2* morpholino exasperated the *wnt11* morpholino midfacial/eye fusion defect (*Waxman et al., 2004*). We hypothesized that the shared phenotypes between *wnt11f2* and *dact1/2* mutants point to these genes acting in the same signaling pathway. To test for genetic epistasis between *wnt11f2*, *dact1*, and *dact2* genes we generated *wnt11f2/dact1/2*−/− triple homozygous mutants. If *wnt11f2* and *dact1/2* had independent developmental requirements, the *wnt11f2/dact1/2*−/− mutant may exhibit a phenotype distinct from *wnt11f2*−/− or *dact1/2*−/− mutants. We found that the *wnt11f2/dact1/2*−/− triple homozygous mutant phenotype of the linear rod-like EP was the same as the *wnt11f2*−/− mutant or *dact1/2*−/− double mutant, without exhibiting additional or neo-phenotypes in the craniofacial cartilages or body axis (*Figure 3D, E*). This result supports *dact1 and dact2* acting downstream of *wnt11f2* signaling during ANC morphogenesis, where loss of *dact1/2* function recapitulates a loss of *wnt11f2* signaling.

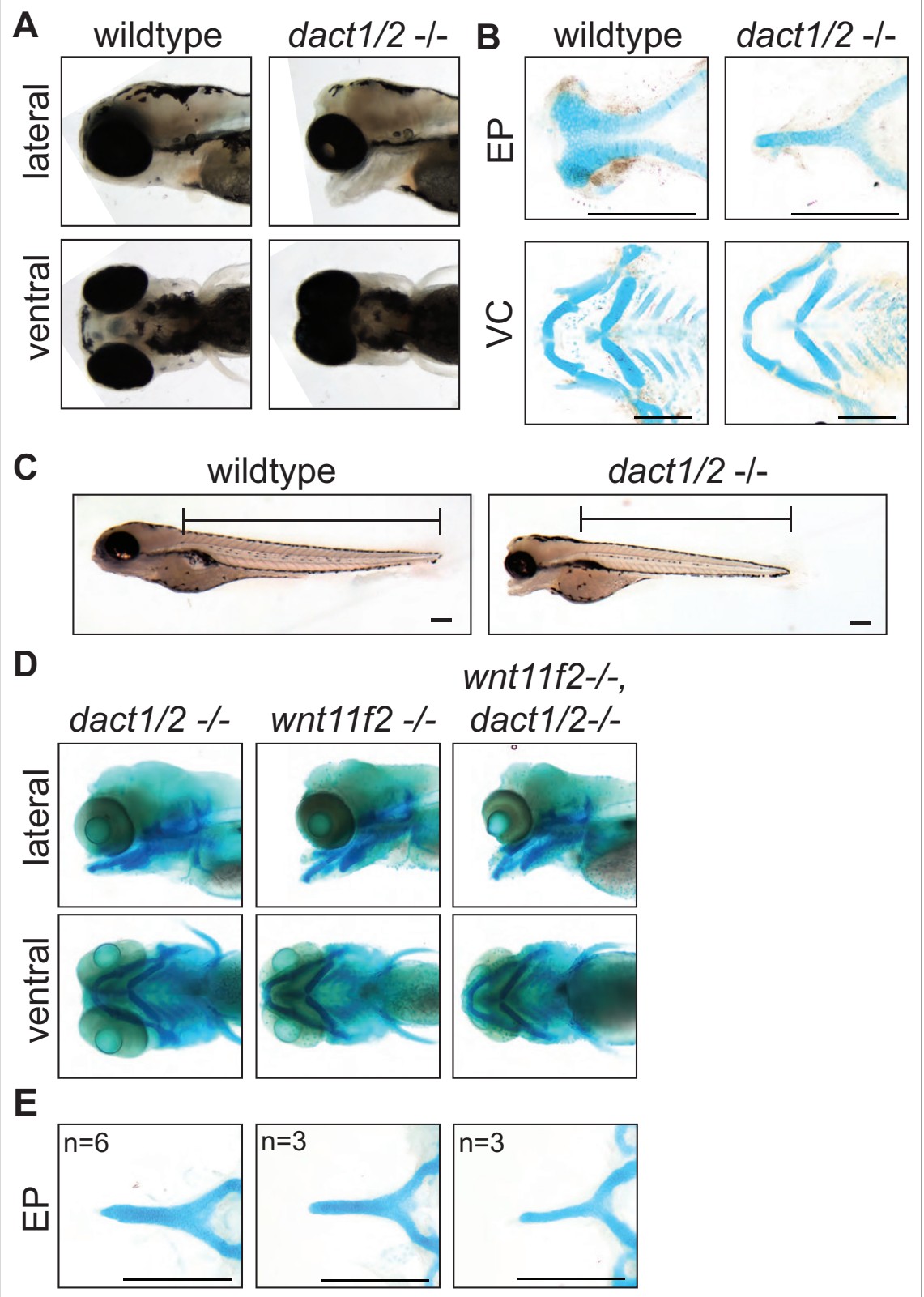

**Figure 3.** Midface development requires *dact1* and *dact2*. (**A**) Representative brightfield images of wildtype and *dact1/2−/−* compound mutants at 4 dpf. 100 individuals were analyzed from a *dact1/2*+/- double het cross. Lateral and ventral views show d*act1/2−/−* compound mutants have a hypoplastic midface, medially displaced eyes, and a displaced lower jaw. (**B**) Representative flat-mount images of Alcian blue stained ethmoid plate (EP) and visceral cartilage (VC) elements from 4 dpf wildtype and d*act1/2−/−* compound mutants. d*act1/2−/−* mutants have a rod-shaped EP with no distinct

*Figure 3 continued on next page*

*Figure 3 continued*

lateral and medial elements. No obvious differences were found in *dact1/2* mutant VC. (**C**) Representative brightfield image of 4 dpf wildtype and *dact1/2−/−* mutant. Bar indicates vertebral spine length. Scale bar: 100 µm. (**D**) Representative images of Alcian blue stained *dact1/2−/−*, *wnt11f2−/−*, and *wnt11f2−/−,dact1/2−/−* compound mutants. Embryos resulted from a *dact1+/-,dact2+/-,wnt11f2+/-*-in-cross. Lateral and ventral views show similar craniofacial phenotypes in each mutant. (**E**) Representative flat-mount images of Alcian blue stained EP show a similar phenotype between *dact1/2−/−*, *wnt11f2−/−*, and *wnt11f2−/−,dact1/2−/−* compound mutants. Scale bar: 200 µm.

## Lineage tracing of *dact1/2* mutant neural crest cell movements reveals their ANC composition

The EP forms from the convergence of a central frontal prominence-derived structure with bilateral maxillary prominence-derived elements (*Wada et al., 2005*; *Swartz et al., 2011*; *Dougherty et al., 2012*; *Mork and Crump, 2015*; *Rochard et al., 2016*).The stereotypic convergent migration of cranial neural crest cells (NCCs) and their derivatives presents an excellent model to examine CE movements and their effects on tissue morphogenesis. The zebrafish EP is formed from the joining of a midline frontal prominence derived from the anteromost cranial NCC population that migrate over the eyes and turn caudally, to join paired lateral maxillary prominences derived from the second stream of cranial NCC population that migrate rostrally (*Kimmel et al., 2001*; *Wada et al., 2005*; *Schilling and Le Pabic, 2009*; *Dougherty et al., 2012*; *Mork and Crump, 2015*). The EP that forms is a planar fan-shaped structure where we and others have shown that the morphology is governed by Wnt signaling (*Kimmel et al., 2001*; *Rochard et al., 2016*).

Given the rod-like EP we observed in the *dact1/2−/−* double mutants, we hypothesized that the dysmorphology could be due to aberrant migration of the anteromost midline stream of cranial NCCs resulting in fusion of the lateral maxillary components. Conversely, an abrogated contribution from the second paired stream of maxillary NCCs could lead to an EP composed entirely of the medial

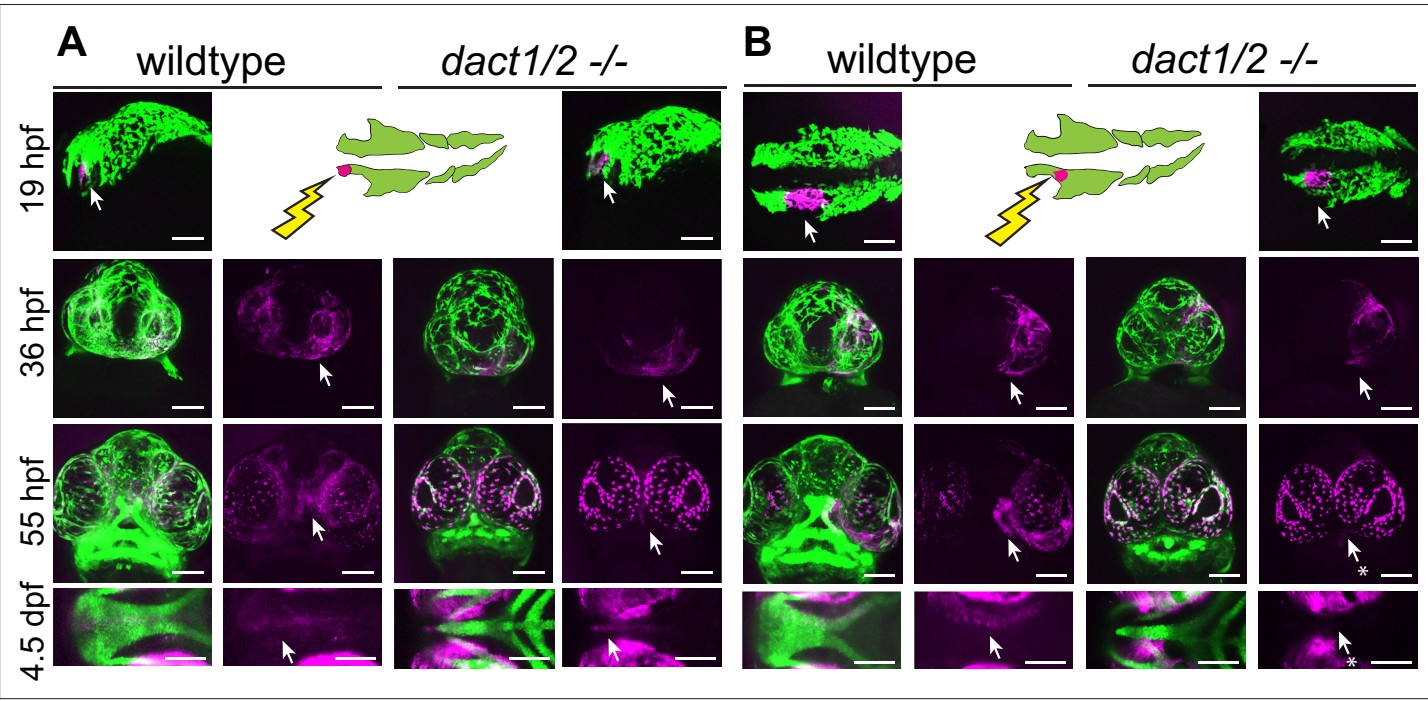

**Figure 4.** Anterior neural crest cells of the *dact1/2−/−* mutant migrate to the midline and populate the dysmorphic ethmoid plate. Lineage tracing of wildtype and *dact1/2−/−* double mutant zebrafish embryos using Tg(*sox10*:kaede) line. *sox10*:kaede fluorescence is shown in green and photo-converted kaede is shown in magenta and highlighted with an arrow. Asterisks indicate that the cell population is absent. (**A, B**) 19 hpf embryo sagittal views showing photoconversion of anterior-most neural crest population. At 36 hpf frontal images show the migration of photoconverted neural crest cells to the frontal prominence in wildtype and *dact1/2−/−* double mutants. At 55 hpf, frontal images show photoconverted neural crest cells populating the region of the developing anterior neurocranium (ANC) in wildtype and *dact1/2−/−* mutants. At 4.5 dpf ventral images show photoconverted neural crest cells populating the medial ethmoid plate in wildtype. Similarly, neural crest cells in *dact1/2−/−* mutants populate the rod-shaped ethmoid plate. Scale bar: 100 µm. Representative images of three individual experiments.

component. To distinguish between these possibilities, we carried out lineage tracing of the cranial NCC populations in wildtype and *dact1/2* mutants. The *dact1/2* compound mutants were bred onto a *sox10:kaede* transgenic background, where we and others have shown that the *sox10* reporter is a reliable driver of cranial neural crest labeling (*Wada et al., 2005*; *Dutton et al., 2008*; *Swartz et al., 2011*; *Dougherty et al., 2012*; *Kague et al., 2012*; *Mork and Crump, 2015*). Cranial NCC populations in wildtype and *dact1/2* mutants were targeted at 19 hpf to photoconvert Kaede reporter protein in either the anterior cranial NCCs that contribute to the frontal prominence, or the second stream of NCCs that contribute to the maxillary prominence, where the labeled cells were followed longitudinally over 4.5 days of development (*Figure 4*). We found that the anterior NCCs of wildtype embryos migrated antero-dorsally to the eye and populated the medial EP. To our surprise, the ante-rior cranial NCC also migrated to contribute to the median element of the rod-like EP, suggesting the complex anterior then caudal migration of the anterior NCC is not disrupted by *dact1/2* mutation (*Figure 4A*, arrows). This finding is in contrast to lineage tracing in another midline mutant with a similarly shaped rod-like EP, the *syu* (*sonic hedgehog* null) mutant, where the anterior NCCs failed to populate the ANC (*Wada et al., 2005*).

Next, the second stream of NCC population that contribute to the maxillary prominence was labeled, where they migrate and contribute to the lateral element of the EP as expected in the wild-type (*Figure 4B*). When the second stream of cranial NCCs were labeled and followed in the *dact1/2* mutants, the cells were found to migrate normally up to 36 hpf, but did not ultimately populate the EP in the mutant (arrows). These results suggest that NCC migration itself is not regulated by *dact1/2* but that loss of *dact1/2* hinders the second stream of NCCs' ability to populate the ANC by an alter-native means. Further, we have found that a rod-like EP can be formed from two different NCC origins, where in the *dact1/2* mutants the EP is contributed by the anteromost frontonasal NCCs, in contrast to the similar rod-shaped EP of the *syu* mutants that is formed from the more posterior stream of maxillary NCCs (*Wada et al., 2005*).

## Genetic interaction of *dact1/2* with *gpc4* and *wls* to determine facial morphology

Given the role of Dact/dapper as modifiers of Wnt signaling, we hypothesized that genetic interac-tion of *dact1/2* with *wls* and *gpc4* will modify facial morphology. Gpc4 is a glycoprotein that binds Wnt ligands and modulates Wnt signaling. *gpc4* zebrafish mutants have impaired CE which leads to a shortened body axis (*Topczewski et al., 2001*). Wls is a posttranslational modifier of Wnt ligands which promotes their secretion (*Bänziger et al., 2006*; *Bartscherer et al., 2006*). We previously described that these components of the Wnt/PCP pathway (*gpc4* receptor, *wls* intracellular ligand chaperon, and Wnt ligands *wnt9a* and *wnt5b*) are required for craniofacial morphogenesis, where each gene affects particular morphologic aspects of chondrocytes arrangement in the cardinal axis of the ANC and Meckel's cartilage (*Rochard et al., 2016*; *Ling et al., 2017*). Using the EP as a morphologic readout, we examined the genetic interaction of *dact1* and *dact2* with *wls* and *gpc4*. Compound mutants of *dact1*, *dact2*, *gpc4*, or *wls* were generated by breeding the single alleles. Compared to wildtype ANC morphology, abrogation of *gpc4* led to increased width in the transverse axis, but shorter in the antero-posterior axis (*Rochard et al., 2016*). Disruption of *wls* leads to ANC morphology that is also wider in the transverse dimension, but to a lesser degree than observed in *gpc4*. Additionally, in the *wls* mutant, chondrocytes stack in greater layers in the sagittal axis (*Rochard et al., 2016*).

Disruption of *gpc4* or *wls* in addition to *dact1/2* generated EP morphology that contained pheno-typic attributes from each single mutant, so that the resultant ANC morphology represented a novel ANC form. The EP of a triple homozygous *gpc4/dact1/2−/−* mutant was triangular, wider in the trans-verse axis and shorter in the antero-posterior axis compared to the rod-like ANC observed in the *dact1/2−/−* double mutant (*Figure 5A, B*). Similarly, the ANC of a triple homozygous *wls/dact1/2−/−* mutant was in the shape of a rod, shorter in the antero-posterior axis and thicker in the sagittal axis compared to the *dact1/2−/−* double mutant, reflecting attributes of the *wls* mutant (*Figure 5C, D*). In addition to the EP phenotypes, the triple homozygous *gpc4/dact1/2−/−* mutant also had a short body axis and truncated tail similar but more severe than the *gpc4* mutant (*Figure 5A*). Since compound disruption of *dact1*, *dact2*, and *gpc4* or *wls* resulted in a new phenotype we conclude that these genes function in different components of Wnt signaling during craniofacial development.

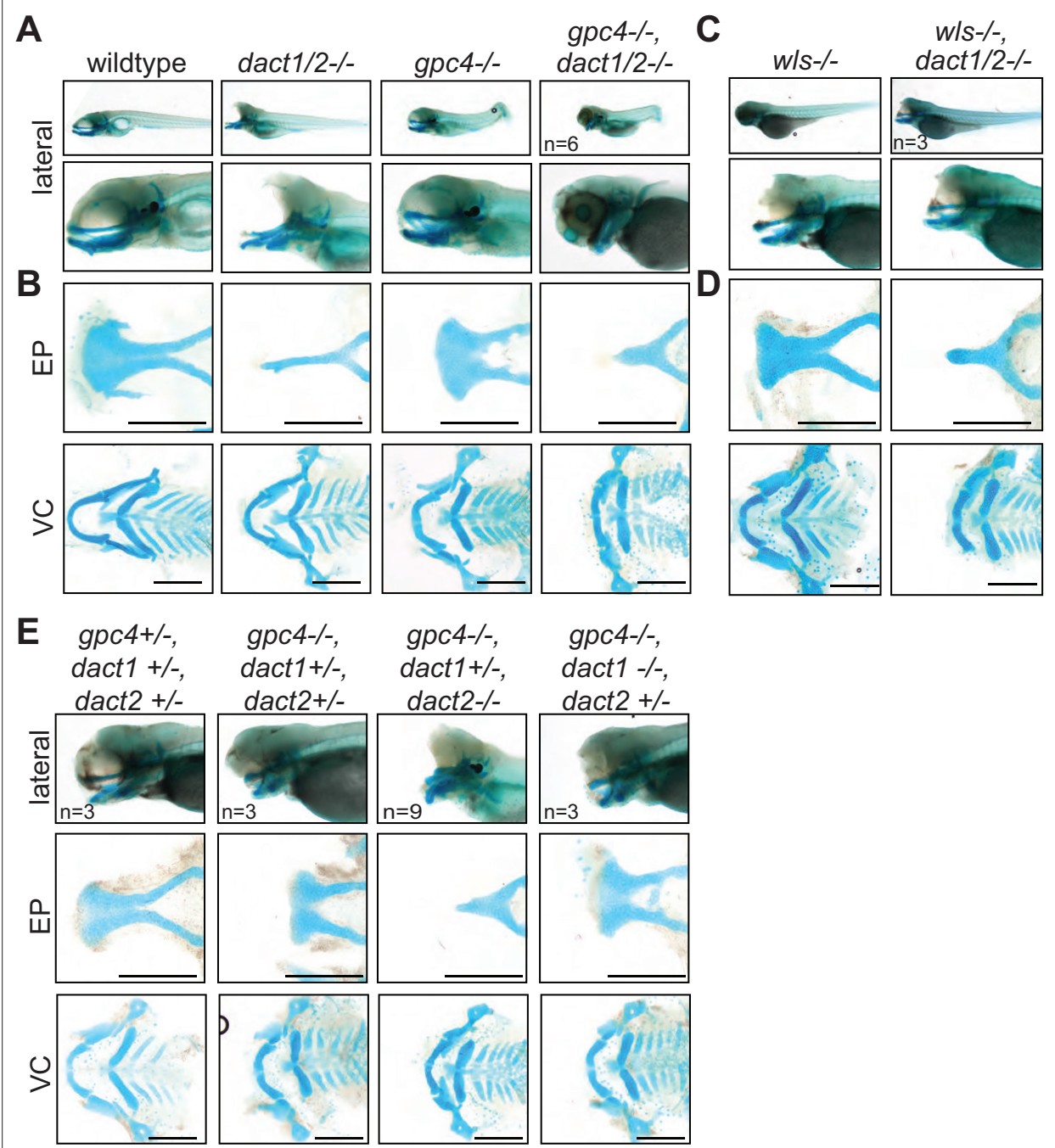

**Figure 5.** A nonoverlapping functional role for *dact1*, *dact2*, and *gpc4* and *wls*. (**A**) Representative Alcian blue stained wholemount images of wildtype, *dact1/2−/−* double mutant, *gpc4−/−* mutant, and *gpc4/dact1/2−/−* triple mutants at 4 dpf. Low magnification lateral images of embryos showing tail truncation in *dact1/2−/−* mutants, shortened and kinked tail in *gpc4−/−* mutants, and a combinatorial effect in *gpc4/dact1/2−/−* triple mutants. Higher magnification lateral images show a shortened midface and displaced lower jaw in *dact1/2−/−* mutants, a shortened midface in *gpc4−/−* mutant, and a combinatorial effect in *gpc4/dact1/2−/−* triple mutants. (**B**) Representative flat-mount images of dissected Alcian blue-stained cartilage elements. *dact1/2−/−* mutants have a narrow rod-shaped ethmoid plate (EP) while *gpc4−/−* mutants have a broad and shortened EP. *dact1/2/gpc4* triple mutants have a combinatorial effect with a short, broad rod-shaped EP. In ventral cartilages (VC), *dact1/2−/−* mutants have a relatively normal morphology while Meckel's cartilage in *gpc4−/−* mutants and *gpc4/dact1/2−/−* triple mutants is truncated. (**C, D**) Same as above except *wls−/−* mutant and *wls/dact1/2−/−* triple mutant, with similar findings. (**E**) Combinatorial genotypes of *dact1*, *dact2*, and *gpc4*. *dact2−/−* contributed the *dact/gpc4* compound phenotype while *dact1−/−* did not. Scale bar: 200 µm.

As we analyzed the subsequent genotypes of our *dact1/dact2/gpc4* triple heterozygote in-cross we gleaned more functional information about *dact1* and *dact2*. We found that *dact1* heterozygosity in the context of *dact2−/−; gpc4−/−* was sufficient to replicate the triple *dact1/dact2/gpc4* homozygous phenotype (*Figure 5E*). In contrast, *dact2* heterozygosity in the context of *dact1−/−; gpc4−/−* double mutant produced ANC in the opposite phenotypic spectrum of ANC morphology, appearing similar to the *gpc4−/−* mutant phenotype (*Figure 5E*). These results show that *dact1* and *dact2* do not have redundant function during craniofacial morphogenesis, and that *dact2* function is more indispensable than *dact1*. These results also suggest that *dact1* and *gpc4* may have overlapping roles in craniofacial development.

## *dact1/2* and *gpc4* regulate axis extension via overlapping and distinct cellular pathways

Our analyses of axis extension and the hallmarks of a CE defect (namely decreased length and increased width between early tissues) demonstrate that *dact1* and/or *dact2* are required for CE and anterior–posterior axis lengthening during gastrulation (*Figure 3*). An axis lengthening and CE defect has also been described in *gpc4* (aka kny) mutants (*Topczewski et al., 2001*). We also observe a defect in axis lengthening in *gpc4−/−* in our hands (representative image *Figure 6A*) that is grossly similar to the *dact1/2−/−* mutants. Interestingly, the midfacial hypoplasia of the *wnt11f2* (slb) mutant has been attributed to a defect in axis extension and anterior neural plate patterning (*Heisenberg and Nüsslein-Volhard, 1997*), whereas defective axis extension does not lead to midfacial hypoplasia in the *gpc4−/−* mutant. Therefore, we hypothesized that by comparing and contrasting the gene expression changes in *dact1/2* versus *gpc4* mutants during axis extension we could identify cell programs specifically responsible for the anterior axis defect and subsequent midfacial hypoplasia. We performed single-cell transcriptional analysis to compare *dact1/2* mutants, *gpc4* mutants, and wildtype embryos during the segmentation stage. Single-cell encapsulation and barcoded cDNA libraries were prepared from individual dissociated 4 ss wildtype, *dact1/2−/−* compound mutant and *gpc4−/−* mutant embryos using the 10X Genomics Chromium platform and Illumina next-generation sequencing. Genotyping of the embryos was not possible but quality control analysis by considering the top 2000 most variable genes across the dataset showed good clustering by genotype, indicating the reproducibility of individuals in each group. Twenty clusters were identified using Louvain clustering and identity was assigned by reviewing cluster-specific markers in light of published expression data (*Farrell et al., 2018*; *Farnsworth et al., 2020*; *Bradford et al., 2022*; *Figure 6B, C*). Qualitatively, we did not observe any significant difference in cluster abundance between genotype groups (*Figure 6—figure supplement 1*). We found that *dact1*, *dact2*, and *gpc4* were detected at various levels across clusters, though *dact1* expression was lower than *dact2* (*Figure 6D*), consistent with what we observed in RNA wholemount ISH analysis (*Figure 1*).

To assess the relative differences in gene expression between genotype groups, we merged clusters into broader cell lineages: ectoderm, axial mesoderm, and paraxial mesoderm (*Figure 7A*). We focused on these cell types because they contribute significantly to CE processes and axis establishment. For each of these cell lineages, we performed independent pseudo-bulk differential expression analyses (DEA) of wildtype versus *dact1/2−/−* mutant and wildtype vs. *gpc4−/−* mutant (*Figure 7A*). In all three cases, we found differentially expressed genes (DEGs) that were commonly in *dact1−/−;dact2−/−* and *gpc4−/−* mutant relative to wildtype (*Figure 7B*). To address the hypothesis that *dact1* and *dact2* regulate molecular pathways distinct from those regulated by *gpc4* we also identified genes that were differentially expressed only in *dact1/2−/−* mutants or only in *gpc4−/−* mutants (*Figure 7B*). Functional analysis of these DEGs found unique enrichment of intermediate filament genes in *gpc4−/−* whereas *dact1/2−/−* mutants had enrichment for pathways associated with proteolysis (*Figure 7C*, *Figure 7—figure supplement 1*). Enrichment for pathways associated with calcium-binding were found in both *gpc4−/−* and *dact1/2−/−*, although the specific DEGs were distinct (*Figure 7—figure supplement 1*). We performed functional analyses specifically for genes that were differentially expressed in *dact1/2−/−* mutants, but not in *gpc4−/−* mutants, and found enrichment in pathways associated with proteolysis (*Figure 7C*) suggesting a novel role for Dact in embryogenesis.

Interrogation of *dact1/2−/−* mutant-specific DEGs found that the calcium-dependent cysteine protease *calpain 8* (*capn8*) was significantly overexpressed in *dact1/2−/−* mutants in paraxial mesoderm (103-fold), axial mesoderm (33-fold), and in ectoderm (3-fold; *Figure 7A*). We also found that

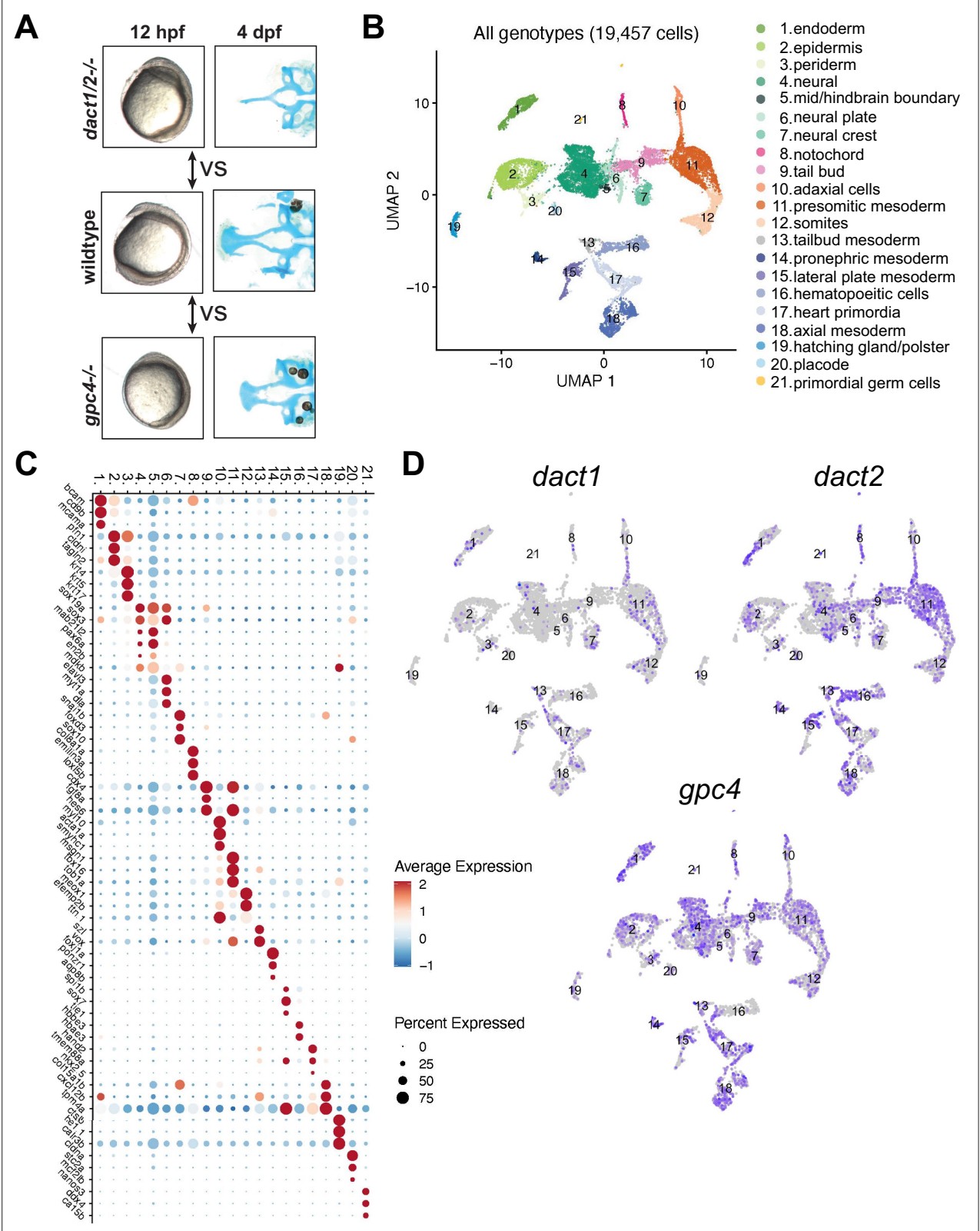

**Figure 6.** Single-cell RNAseq of 4 ss wildtype, *dact1/2−/−* mutant, and *gpc4−/−* mutants. (**A**) Summary schematic showing similar phenotypes in *dact1/2−/−* and *gpc4−/−* mutants at 12 hpf and divergent phenotypes at 4 dpf. Single-cell RNAseq was performed during axis extension to compare and contrast *dact1/2−/−* and *gpc4−/−* transcriptional programs. Uniform manifold approximation and projection (UMAP) showing cluster identification.

*Figure 6 continued on next page*

*Figure 6 continued*

(**B**) UMAP of cell clusters identified by single-cell RNAseq. (**C**) Dot plot showing the most differentially expressed genes between clusters. (**D**) UMAP showing *dact1*, *dact2*, and *gpc4* expression in wildtype embryos.

The online version of this article includes the following figure supplement(s) for figure 6:

**Figure supplement 1.** Cluster abundance across genotype groups.

loss of *dact1/2* causes significant changes to *capn8* expression pattern (**Figure 8A**). Whereas *capn8* gene expression is principally restricted to the epidermis of wildtype embryos, loss of *dact1/2* leads to significant expansion of ectopic *capn8* gene expression in broader cell types such as in mesodermal tissues (**Figure 8A**). We corroborated this finding with wholemount RNA ISH for *capn8* expression in wildtype versus *dact1/2−/−* 12 hpf embryos (**Figure 8B**). The expression of *smad1* was found to be decreased uniquely in the ectoderm of *dact1/2−/−* embryos relative to wildtype (**Figure 7A**), however this finding was not investigated further.

*Capn8* is considered a 'classical' calpain, with domain homology similar to *Capn1* and *Capn2* (**Macqueen and Wilcox, 2014**). In adult human and mouse tissue, *Capn8* expression is largely restricted to the gastrointestinal tract (**Sorimachi et al., 1993**; **Macqueen and Wilcox, 2014**), however embryonic expression in mammals has not been characterized. Proteolytic targets of Capn8 have not been identified, however, other classical calpains have been implicated in Wnt and cell–cell/ECM signaling (**Konze et al., 2014**), including in Wnt/Ca$^{+2}$ regulation of CE in *Xenopus* (**Zanardelli et al., 2013**). To determine whether the *dact1/2−/−* mutant craniofacial phenotype could be attributed to *capn8* overexpression, we performed injection of *capn8* or *gfp* control mRNA into 1 cell-stage zebrafish embryos. In wildtype zebrafish, exogenous *capn8* mRNA caused the distinct *dact1/2−/−* craniofacial phenotype including a rod-like ANC at a very low frequency (1 in 142 injected embryos). This craniofacial phenotype was not observed in wildtype larvae, or when wildtype embryos were injected with an equal concentration of *gfp* mRNA (0 in 192 injected embryos) (data not shown). When mRNA was injected into 1 cell-stage embryos generated from *dact1/2+/-* interbreeding, *capn8* caused a significant increase in the number of larvae with the mutant craniofacial phenotype when on a *dact1/2+/-* genetic background (**Figure 8D**, 0.0% vs. 7.5%). We did not find an effect of exogenous *capn8* on any other genotype, including *dact1−/−,dact2+/-* which we suspect to be due to the smaller number of those individuals in our experimental population. These findings suggest a new contribution of capn8 to embryonic development as well as anterior neural plate patterning and craniofacial development. Further, the regulation of *capn8* by *dact* may be required for normal embryogenesis and craniofacial morphogenesis.

## Discussion

In this study, we examined the genetic requirement of *dact1* and *dact2* during early embryogenesis and craniofacial morphogenesis in zebrafish. Wnt signaling is central to the orchestration of embryogenesis and numerous proteins have been identified as modulators of Wnt signaling, including *Dact1* and *Dact2* (**Cheyette et al., 2002**). Several studies across *Xenopus*, zebrafish, and mouse have ascribed roles to *dact1* and *dact2*, including both promoting and antagonizing Wnt signaling, depending on the developmental context (**Cheyette et al., 2002**; **Gloy et al., 2002**; **Waxman et al., 2004**; **Gao et al., 2008**; **Wen et al., 2010**; **Ma et al., 2015**; **Lee et al., 2018**). Here, we show that *dact1* and *dact2* are required for axis extension during gastrulation and show an example of CE defects during gastrulation associated with craniofacial defects. During axis extension, we show that genetic disruption of *dact2*, but not *dact1*, resulted in a significantly shortened axis relative to wildtype. This result is similar to what was previously found using morpholinos to disrupt *dact1* and *dact2*. Interestingly, genetically disrupted mutants of *dact1* or *dact2* developed to be phenotypically normal whereas *dact1/2* compound mutants displayed a severe dysmorphic craniofacial phenotype. Again, this is largely similar to the previous morpholino study that found disruption of each gene to cause only a slight and occasional dysmorphic cranial phenotype at 24 hpf (**Waxman et al., 2004**). Notably, embryos injected with a mixture of *dact1* and *dact2* morpholino were not characterized after 10 ss, and the singly injected embryos were not characterized after 24 hpf (**Waxman et al., 2004**). Therefore, by analyzing genetic mutants of dact1 and dact2 our findings have largely validated the previous morpholino literature as well as added new data on later developmental outcomes.

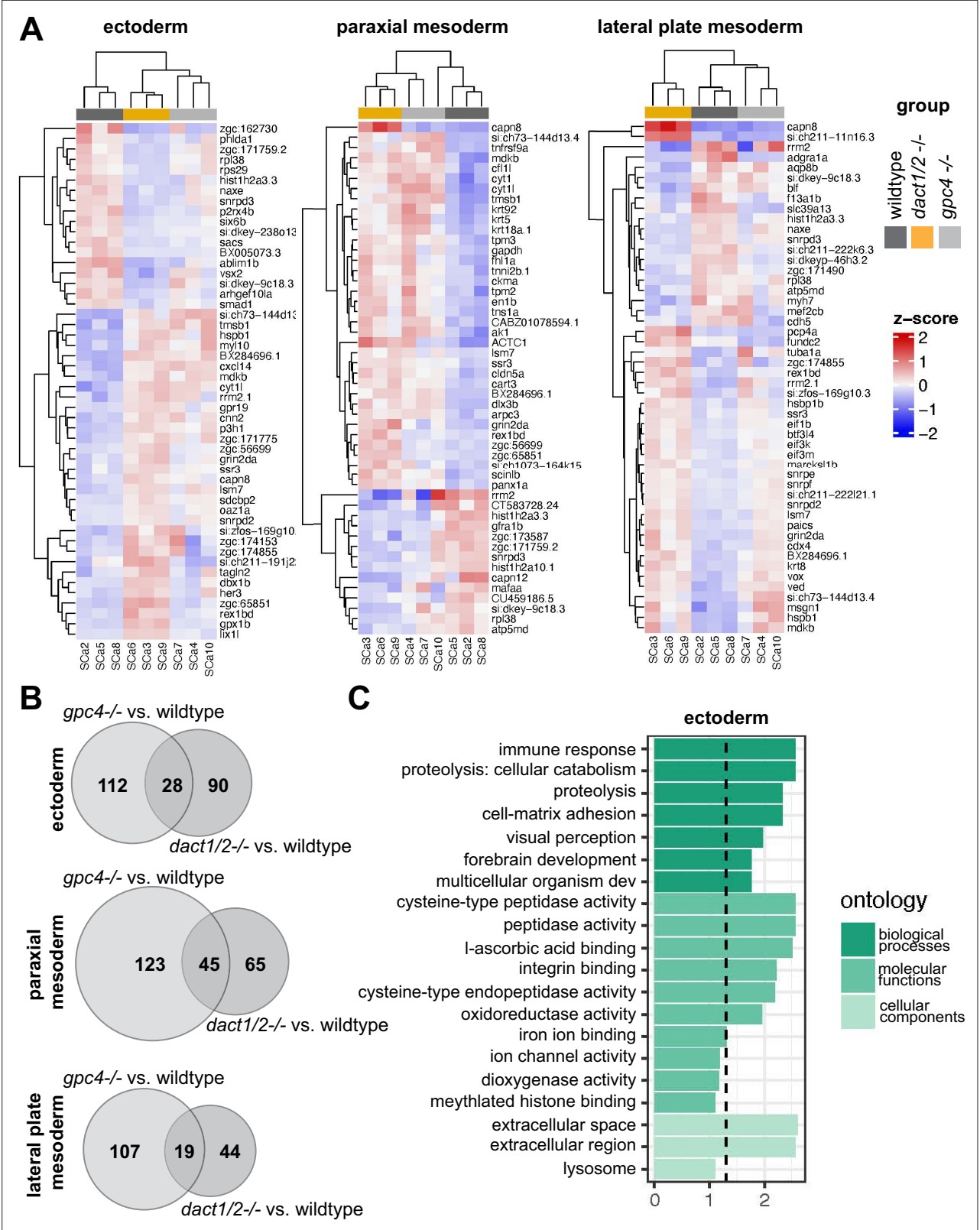

**Figure 7.** Pseudobulk differential expression analysis of single-cell RNAseq data. (**A**) Heatmaps showing the 50 most differentially expressed genes (DEGs) in 3 major cell types; ectoderm (clusters 4, 5, 6, 7), paraxial mesoderm (clusters 10, 11, 12), and lateral plate mesoderm (clusters 15, 16, 17,18) between *dact1/2−/−* mutants and wildtype and *gpc4−/−* mutants and wildtype. (**B**) Venn diagrams showing unique and overlapping DEGs

*Figure 7 continued on next page*

*Figure 7 continued*

in *dact1/2−/−* and *gpc4−/−* mutants. (**C**) Gene Ontology (GO) analysis of *dact1/2−/−* mutant-specific DEGs in ectoderm showing enrichment for proteolytic processes.

The online version of this article includes the following figure supplement(s) for figure 7:

**Figure supplement 1.** Loss of *gpc4* and loss of *dact1/2* lead to distinct changes in gene expression profiles but with some overlapping functions.

The gene expression and genetic epistasis experiments carried out here support that the *dact* paralogs are not redundant and have unique functions during different stages of embryonic and larval development. We observed that *dact1* and *dact2* have distinct spatiotemporal expression patterns throughout embryogenesis, suggesting unique roles for each paralog in developmental processes. Differential expression of *Dact1* and *Dact2* was also described during odontogenesis in mice (**Kettunen et al., 2010**). This aligns with previous findings of differential roles of *dact1* and *dact2* in canonical versus noncanonical Wnt signaling (**Waxman et al., 2004**) and a specific role for *dact2*, but not *dact1*, in TGF-β signaling (**Su et al., 2007**; **Schubert et al., 2014**). However, the lack of a resultant phenotype upon genetic ablation of *dact1* or *dact2* individually suggests the capability of functional compensation. This is puzzling given their distinct expression patterns and needs to be examined further.

We found that *dact1* and *dact2* contribute to axis extension, and their compound mutants exhibit a shortened and widened body axis that is consistent with a CE defect during gastrulation. This finding aligns with previous studies that have implicated *dact1* and *dact2* in noncanonical Wnt signaling and regulation of embryonic axis extension (**Waxman et al., 2004**). Based on our gene expression and combinatorial genetic analyses, we offer the hypothesis that *dact1* expression in the paraxial meso-derm is required for dorsal CE during gastrulation through its role in noncanonical Wnt/PCP signaling, similar to the defect observed upon *gpc4* disruption. Conversely, we posit that *dact2* functions in the prechordal mesoderm to promote anterior migration during gastrulation, a function which has also been ascribed to *wnt11f2* (**Heisenberg and Nüsslein-Volhard, 1997**). It is only upon loss of both *dact1* and *dact2* functions that the axis is significantly truncated and a craniofacial malformation results. Further experiments with spatially restricted gene ablation or cell transplantation are required to test this hypothesis.

Our results underscore the crucial roles of *dact1* and *dact2* in embryonic development and suggest a connection between gastrulation movements and subsequent craniofacial morphogenesis. Our finding that in *dact1−/−;dact2−/−* compound mutants the first stream of cranial NCC migrate and contribute to the ANC, while the second stream fails to contribute suggests the possibility of an anatomical barrier to migration, rather than an autonomous defect of the cranial NCCs. Disruption of the sonic hedgehog signaling pathway in zebrafish results in a similar phenotype to *dact1/2−/−* and *wnt11f2−/−* mutants where the eyes converge medially and the EP narrows to a rod shape. Interest-ingly, lineage tracing analysis in hedgehog-disrupted embryos found the rod-like EP to consist solely of second stream-derived cranial NCCs (**Wada et al., 2005**). This is in contrast to the *dact1/2−/−* mutants, demonstrating two different cellular mechanisms that result in a similar anatomical dysmor-phology. It will be important to test the generality of this phenomenon and determine if other mutants with craniofacial abnormalities have early patterning differences. Further, a temporally conditional genetic knockout is needed to definitively test the connection between early and later development.

By comparing the transcriptome across different Wnt genetic contexts, that is *gpc4−/−* with that of the *dact1/2−/−* compound mutant, we identified a novel role for *dact1/2* in the regulation of prote-olysis, with significant misexpression of *capn8* in the mesoderm of *dact1/2−/−* mutants. Although at a very low frequency, ectopic expression of *capn8* mRNA recapitulated the *dact1/2−/−* mutant cranio-facial phenotype, suggesting that inhibition of *capn8* expression in the mesoderm by dact is required for normal morphogenesis. Genes involved in calcium ion binding were also differentially expressed in the *dact1/2−/−* mutants and we predict that altering intracellular calcium handling in conjunction with *capn8* overexpression would increase the frequency of the recapitulated *dact1/2−/−* mutant phenotype.

Capn8 is described as a stomach-specific calpain and a role during embryogenesis has not been previously described. Calpains are typically calcium-activated proteases and it is feasible that Capn8 is active in response to Wnt/Ca$^{2+}$ signaling. A close family member, Capn2 has been found to modu-late Wnt signaling by degradation of beta-catenin (**Zanardelli et al., 2013**; **Konze et al., 2014**). Our findings suggest that dact-dependent suppression of *capn8* expression is necessary for normal

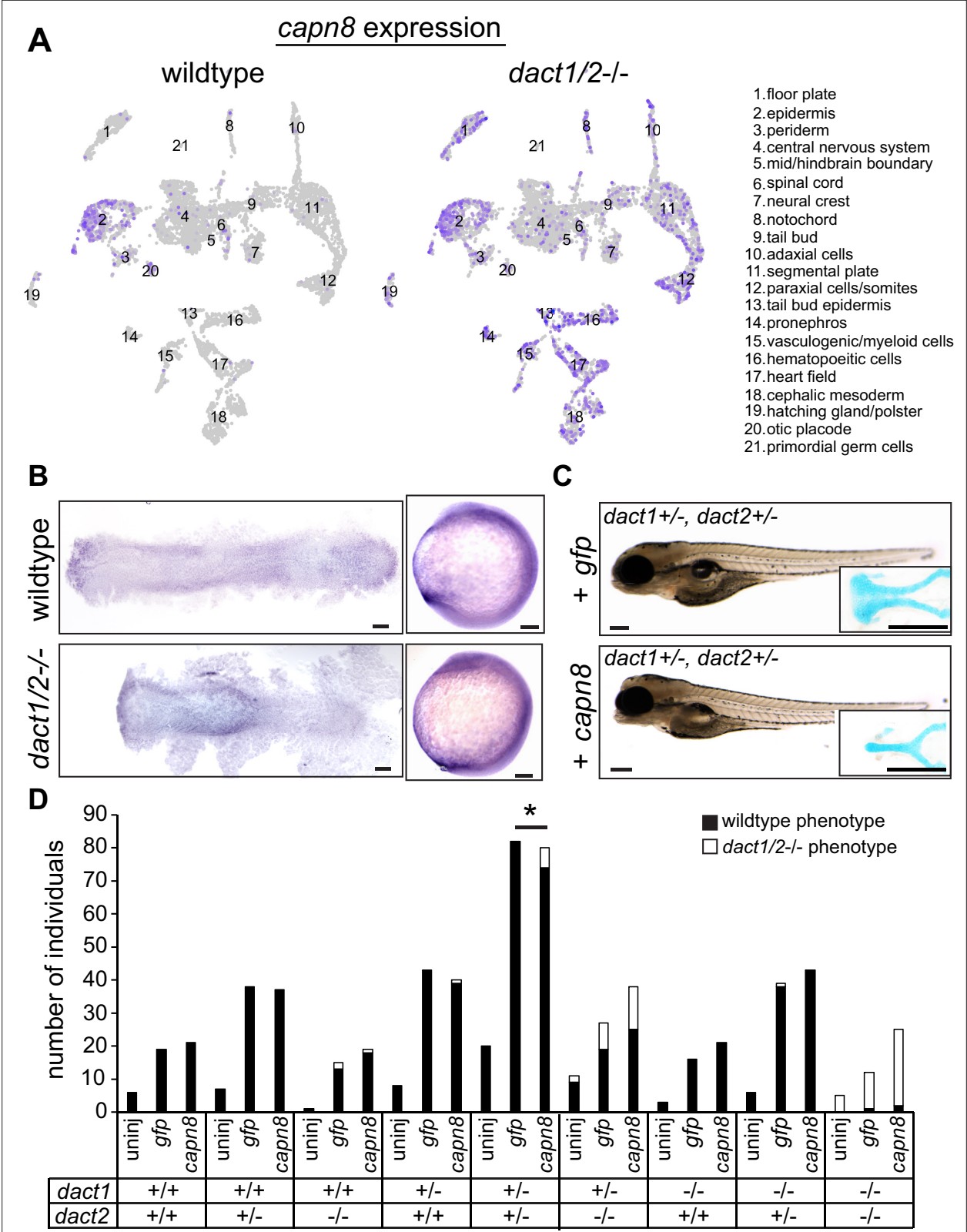

**Figure 8.** Expression of *capn8* is significantly dysregulated in *dact1/2−/−* mutants. (**A**) Single-cell RNAseq gene expression analysis of *capn8* in wildtype and *dact1/2−/−* mutants. In wildtype embryos, *capn8* expression is restricted predominantly to the epidermis whereas *capn8* is widely expressed throughout the embryo in *dact1/2−/−* mutants, especially in the mesoderm. (**B**) Wholemount in situ hybridization of *capn8* expression in wildtype and *dact1/2−/−* mutant embryos at 2 ss. Staining corroborates the single-cell RNAseq data, with expanded ectopic expression of *capn8* throughout

*Figure 8 continued on next page*

*Figure 8 continued*

the embryo. Flat mounts are oriented anterior to the left. Scale bar: 100 μm. (**C**) Brightfield images and Alcian blue staining of the ethmoid plate show ectopic expression of *capn8* mRNA (200 pg) at the 1 cell stage in *dact1+/-,dact2+/-* embryos recapitulates the *dact1/2−/−* compound mutant craniofacial phenotype. The mutant craniofacial phenotype did not manifest in gfp mRNA (200 pg) injected 1 cell-stage *dact1+/-,dact2+/-* embryos. Scale bar: 100 μm (**D**) Quantification of mutant and normal craniofacial phenotype in 4 dpf larvae after mRNA injection at the 1 cell stage. Larvae were derived from *dact1/2+/-* interbreeding. Larvae were uninjected or injected with 200 pg *gfp* control or *capn8* mRNA. A Fisher exact test showed a significant effect of *capn8* mRNA injecting in the *dact1/2* double heterozygotes. Asterisk indicates a significant difference between conditions (p = 0.013).

embryogenesis and craniofacial morphogenesis, further expanding the functional repertoire of *dact1/2*. Continued research is required to test a direct regulatory role of dacts on *capn8* expression. While our data suggests an interaction between dact signaling and capn8 function, we did not find capn8 overexpression to be wholly sufficient to cause the rod-like EP phenotype, Further, we did not test the necessity of *capn8* for craniofacial development in this study. This study does however identify capn8 as a novel embryonic gene warranting further investigation into its role during embryogenesis, with possible implications for known craniofacial or other disorders. Recently, Capn8 has been implicated in EMT associated with cancer metastasis (*Zhong et al., 2022*; *Song et al., 2024*) and *Xenopus* capn8 was found to be required for cranial NCC migration (*Cousin et al., 2011*) which further supports a role of *capn8* in cranial NCC migration and craniofacial morphogenesis.

Another gene identified in our single-cell RNA sequencing data to be differentially expressed in the *dact1/22−/−* but not the *gpc4−/−* embryos was *smad1*. Smad1 acts in the TGF-β signaling pathway and dact2 has been described to inhibit TGF-β and Nodal signaling by promoting the degradation of Nodal receptors (*Zhang et al., 2004*; *Su et al., 2007*; *Meng et al., 2008*; *Kivimäe et al., 2011*). Zebrafish Nodal pathway mutants (*cyc/ndr2*, *oep/tdgf1*, *sqt/ndr1*) exhibit medially displaced eyes (*Hatta et al., 1991*; *Brand et al., 1996*; *Heisenberg and Nüsslein-Volhard, 1997*; *Feldman et al., 1998*; *Zhang et al., 1998*) and it is robustly feasible that dysregulation of TGF-β signaling in the *dact1/2−/−* mutant contributes to the craniofacial phenotype. Future research will examine the role of *dact1* and *dact2* in the coordination of Wnt and TGF-β signaling and the importance of this coordination in the context of craniofacial development. Of note, Sonic Hedgehog (shh) signaling is a principal regulator to the vertebrate midline (*Chiang et al., 1996*; *Ribes et al., 2010*), and important in the development of the zebrafish floorplate (*Halpern et al., 1997*; *Odenthal et al., 2000*). Mutants with disrupted *shh* expression or signaling (*cyc/ndr2*, *smo*, *oep/tdgf1*) exhibit medially displaced eyes similar to the *dact1/2* mutants (*Brand et al., 1996*; *Chen et al., 2001*). We did not find any genes within the sonic hedgehog pathway to be differentially expressed in *dact1/2* mutants, though post-transcriptional interactions cannot be ruled out.

This study has uncovered the genetic requirement of *dact1* and *dact2* in embryonic CE and craniofacial morphogenesis, delineated the genetic interaction with Wnt genes and identified capn8 as a modifier of this process. Future work will delineate the molecular differences across the different *dact1/2* and other Wnt mutants to further identify determinants of craniofacial morphogenesis; and to connect these findings to clinically important Wnt regulators of facial morphology and pathology.

## Materials and methods

### Key resources table

| Reagent type (species) or resource | Designation | Source or reference | Identifiers | Additional information |
|---|---|---|---|---|
| Strain (*Danio rerio*) | WT (AB) RRID:ZIRC_ZL1 | Zebrafish International Resource Center | ZDB-GENO-960809-7 | |
| Strain (*Danio rerio*) | WT (Tubingen) RRID:NCBITaxon_7955 | Zebrafish International Resource Center | ZDB-GENO-990623-3 | |

*Continued on next page*

*Continued*

| Reagent type (species) or resource | Designation | Source or reference | Identifiers | Additional information |
|---|---|---|---|---|
| Strain (*Danio rerio*) | wnt11f2 RRID:ZFIN_ZDB-GENO-200617-11 | Zebrafish International Resource Center | wnt11f2tx226/+ | |
| Strain (*Danio rerio*) | gpc4 RRID:ZFIN_ZDB-GENO-070209-132 | Zebrafish International Resource Center | gpc4hi1688Tg/+ | |
| Strain (*Danio rerio*) | wls | Gift. ***Rochard et al., 2016*** PMID:27287801 | | |
| Strain (*Danio rerio*) | sox10:kaede | ***Dougherty et al., 2012*** PMID:22948622 | | |
| Strain (*Danio rerio*) | dact1 | This paper | | Methods: Animals and CRISPR/Cas9 targeted mutagenesis |
| Strain (*Danio rerio*) | dact2 | This paper | | Methods: Animals and CRISPR/Cas9 targeted mutagenesis |
| Commercial kit | RNeasy Plus Mini Kit | QIAGEN | ID_source:identifier 74134 | |
| Commercial kit | High Capacity cDNA Reverse Transcription Kit | Thermo Fisher | ID_source:identifier 4368814 | |
| Commercial assay or kit | dact1 gene expression assay | Thermo Fisher | Dr03152516_m1 | |
| Commercial assay or kit | dact2 gene expression assay | Thermo Fisher | Dr03426298_s1 | |
| Commercial assay or kit | 18S rRNA gene expression assay | Thermo Fisher | Hs03003631_g1 | |
| Recombinant DNA reagent | pCS2+8 destination plasmid | Addgene ***Gökirmak et al., 2012*** PMID:23124201 | #34931 | |
| Commercial assay or kit | ImMessage mMachine | Invitrogen | ID_source:identifier AM1344 | |
| Commercial assay or kit | RNAscope probe dact1 | ACDbio | ID_source:identifier 857191-C2 | |
| Commercial assay or kit | RNAscope probe dact2 | ACDbio | ID_source:identifier 857201-C3 | |
| Commercial assay or kit | RNAscope probe irf6 | ACDbio | ID_source:identifier 555101 | |
| Commercial assay or kit | Chromium Single Cell 3' kit (version 3) | 10X Genomics | ID_source:identifier 1000268 | |
| Software, algorithm | Cellranger (version 6.1.0) 10x Genomics Cellranger DNA (RRID:SCR_023221) | ***Zheng et al., 2017*** PMID:28091601 | | |
| Software, algorithm | Seurat (version 4.1.0) SEURAT (RRID:SCR_007322) | ***Hao et al., 2021*** PMID:34062119 | | |
| Software, algorithm | Harmony (version 0.1.0) Harmony (RRID:SCR_022206) | ***Korsunsky et al., 2019*** PMID:31740819 | | |
| Software, algorithm | DESeq2 (v1.34.0) DESeq2 (RRID:SCR_015687) | ***Love et al., 2014*** PMID:25516281 | | |
| Software, algorithm | clusterProfiler (version 4.2.2) clusterProfiler (RRID:SCR_016884) | ***Wu et al., 2021*** | | |
| Software, algorithm | ZiFiT Targeter v4.2 | ***Sander et al., 2007*** PMID:17526515 | | |
| Software, algorithm | ChopChop CHOPCHOP (RRID:SCR_015723) | ***Montague et al., 2014*** PMID:24861617 | | |

## Animals and CRISPR/Cas9 targeted mutagenesis

All animal husbandry and experiments were performed in accordance with and approval from the Massachusetts General Hospital Institutional Animal Care and Use Committee (protocol number 2010N000106) and the Children's Hospital of Philadelphia Animal Care and Use Committee (protocol number IAC22001475). Zebrafish (*Danio rerio*) embryos and adults were maintained in accordance with institutional protocols. Embryos were raised at 28.5°C in E3 medium (*Carroll et al., 2020*) and staged visually and according to standardized developmental time points (*Westerfield, 1993*). All zebrafish lines used for experiments and gene editing were generated from the Tubingen or AB strain. The *wnt11f2* mutant line and gpc4−/− mutant line were obtained from Zebrafish International Resource Center (wnt11f2^tx226/+ and gpc4^hi1688Tg/+, respectively). The *wls*−/− mutant line was originally gifted to the lab and independently generated, as previously described (*Rochard et al., 2016*). The *sox10*:kaede transgenic line was previously generated and described by our lab (*Dougherty et al., 2012*).

CRISPR sgRNA guides were designed using computational programs ZiFiT Targeter v4.2 (https://zifit.partners.org/ZiFit) (*Sander et al., 2007*), crispr.mit.edu (https://zlab.bio/guide-design-resouces) (*Ran et al., 2013*), and ChopChop (https://chopchop.cbu.uib.no) (*Montague et al., 2014*) with traditional sequence constraints. Guides were chosen that were predicted to give high efficiency and specificity. Guides best meeting these parameters were selected in exon 2 of *dact1* and exon 4 of *dact2*. No suitable gRNA with sufficient efficiency were identified for *dact2* 5′ of exon 4 and the resulting phenotype was reassuring compared to previous morpholino published results. Guides for *dact1* and *dact2* and Cas9 protein were prepared and microinjected into 1 cell-stage zebrafish embryos and founders were identified as previously described (*Carroll et al., 2020*). Primers flanking the sgRNA guide site were designed for genotyping and fragment analysis was performed on genomic DNA to detect base pair insertion/deletion. Sanger sequencing was performed to verify targeted gene mutation and confirm the inclusion of a premature stop codon. *dact1* forward primer: TACAGAAGCTGCTGAAGTACCG, *dact1* reverse primer: CCCTCTCTCAAAGTGTTTTGGT, *dact2* forward primer: TGAAGAGCTCCACTCCCCTGT, *dact2* reverse primer: GCAGTTGAGGTCCATTCAGC.

## RT-qPCR analysis

Pooled wildtype, *dact1*−/−, *dact2*−/−, and *dact1/2*−/− fish were collected and RNA extractions were performed using RNeasy Mini Kit (QIAGEN). cDNA was generated using High Capacity cDNA Reverse Transcription Kit (ThermoFisher). Quantitative PCR was performed using *dact1* (Dr03152516_m1) and *dact2* (Dr03426298_s1) TaqMan Gene Expression Assays. Expression was normalized to 18S rRNA expression (Hs03003631_g1). TaqMan Fast Advanced master mix (Thermo Fisher) and a StepOnePlus Real-Time PCR system (Applied Biosystems) were used to measure relative mRNA levels, which were calculated using the ddCT method.

## Microinjection of mRNA

Template DNA for in vivo mRNA transcription was generated by PCR amplification of the gene of interest from a zebrafish embryo cDNA library and cloning into pCS2+8 destination plasmid. mRNA for injection was generated using an in vitro transcription kit (Invitrogen mMessage mMachine). One-cell-stage zebrafish embryos were injected with 2 nl of mRNA in solution. To test genetic knockout specificity, 150 or 300 pg or dact1, dact2, or dact1 and dact2 mRNA was injected. For *capn8* overexpression analysis 200 pg or GFP or *capn8* mRNA was injected. Following phenotyping analysis, genotypes were determined by fragment analysis of *dact1* and *dact2* genotyping PCR products.

## Wholemount and RNAscope ISH

Wholemount ISH was performed as previously described (*Carroll et al., 2020*). Zebrafish embryonic cDNA was used as a template to generate riboprobes. Primers were designed to PCR amplify the specific riboprobe sequence with a T7 promoter sequence linked to the reverse primer. In vitro transcription was performed using a T7 polymerase and DIG-labeled nucleotides (Roche). Probe primer sequences are; *dact1* fwd; 5′AGCGCGATTCTCAGATGCAC3′, rev 5′gaaatTAATACGACTCACTATAggCCTGCTCGGGTTTCTGTTCA3′, *dact2* fwd: 5′CAGTCGCATAGCGGATCTCAT3′, rev: 5′gaaatTAATACGACTCACTATAggGTGGACTGGGGTAACGGTAA3′, *wnt11f2* fwd: 5′TCCGTGGTGTATCTTGACCG3′, rev: 5′gaaatTAATACGACTCACTATAggCTTGGTGGCCGACAGGTATT3′, *pax2a* fwd: 5′CCAAACC

AAAAGTGGCGACG3', rev: 5'gaaatTAATACGACTCACTATAggGTTGCTGAACCGCCAAGC3', *gsc* fwd: 5'CCAGCGCCGAACTTACAATC3', rev: 5'gaaatTAATACGACTCACTATAggTCTTCAGCTACAGCC CATTCC3', *zic1* fwd: 5'TAGGGGATCGGAGTTTGCCT3', rev: 5'gaaatTAATACGACTCACTATAggTTC GTCAGCTGCTCTGGTTC3', *tbx6* fwd: 5'ACAGAGATCGAGATGTGCCG3', rev: 5'gaaatTAATACGACT CACTATAggTGGAAGGGCGGTGTTCATAA3', *myo1d* fwd: 5'TCTACGACGACCCTTGCTTC3', rev: 5'gaaatTAATACGACTCACTATAggGGTCGGATTCGCCTTTTTCTG3', *ctslb1* fwd: 5'AGACCGCCTCT ATGTTCGGA3', rev: 5'gaaatTAATACGACTCACTATAggAGCGACATTAAAACGGGGGT3', *capn8* fwd: 5'AAGGGCTGGGGACAAATGAG3', rev: 5'gaaatTAATACGACTCACTATAggCACTAGGAATGTGCA GCCGT3'.

RNAscope was performed on sectioned zebrafish larvae as previously described (*Carroll et al., 2020*). Probes were designed by and purchased from ACD Bio. Hybridization and detection were performed according to the manufacturer's protocol. Sections were imaged using a confocal microscope (Leica SP8) and z-stack maximum projections were generated using Fiji software.

## Alcian blue staining and imaging

Alcian blue staining and imaging were performed at 4 dpf as previously described (*Carroll et al., 2020*). Briefly, larvae were fixed in 4% vol/vol formaldehyde overnight at 4°C. Larvae were dehydrated in 50% vol/vol ethanol and stained with Alcian blue as described (*Walker and Kimmel, 2007*). Whole and dissected larvae were imaged in 3% wt/vol methylcellulose using a Nikon Eclipse 80i compound microscope with a Nikon DS Ri1 camera. Z-stacked images were taken and extended depth of field was calculated using NIS Element BR 3.2 software. Images were processed using Fiji software. After image capture embryos were genotyped by PCR and fragment analysis.

## Axis measurements

Compound *dact1+/-; dact2+/-* zebrafish were in-crossed and progeny were collected from two separate clutches and fixed in 4% formaldehyde at approximately 8 ss. Embryos were individually imaged using a Zeiss Axiozoom stereoscope and processed for DNA extraction and genotyping (*Westerfield, 1993*). Images were analyzed using Fiji. A circle was drawn to overlay the yolk and the geometric center was determined using the function on Fiji. Using the Fiji angle tool, lines were drawn from the center point to the anterior-most point of the embryo and from the center to the posterior-most point of the embryo. The resulting inner angle of these lines was determined. Each angle measurement was then calculated as a ratio to the average angle of the wildtype embryos. All measurements were performed on images blinded for genotype. ANOVA was performed to determine statistical significance, p < 0.05.

## Lineage analysis of cranial NCCs

Live embryos were mounted in 1% wt/vol low melt agarose and covered with E3 medium containing 0.013% wt/vol tricaine. Wildtype control and dact1/2−/− compound mutants on a Tg(*soxl0*:kaede) background were imaged on a Leica DMi8 confocal microscope and photoconverted using the UV laser (404 nm) until the green kaede fluorescence disappeared. For each embryo, one side was photoconverted and the contralateral side served as an internal control. After photoconversion, embryos were removed from the agarose and raised in E3 medium at 28.5°C until the required developmental time point, at which time they were similarly re-mounted and re-imaged. Z-stacked images were processed as maximum-intensity projections using Fiji software.

## Single-cell RNA sequencing

Single-cell transcriptomic analyses were performed on 10 zebrafish embryos, including 4 wildtype, 3 *dact1−/−;dact2−/−* compound mutant, and 3 *gpc4−/−* mutant embryos. Embryos were collected at 4 ss with *dact1−/−,dact2−/−* and *gpc4−/−* mutants being identified by their truncated body axis. Embryos were dechorionated with a short (approximately 10 min) incubation in 1 mg/ml Pronase and then washed 3× in embryo medium. Cell dissociation was performed with modifications as previously described (*Farrell et al., 2018*). Each embryo was transferred to 50 µl DMEM/F12 media on ice. To dissociate cells, media was replaced with 200 µl Dulbecco's phosphate-buffered saline (DPBS) (without $Ca^{2+}$ and $Mg^{2+}$) with 0.1% wt/vol bovine serum albumin (BSA). Embryos were disrupted by pipetting 10× with a P200 pipette tip. 500 µl of DPBS + 0.1% BSA was added and cells were centrifuged at 300

× *g* for 1 min. Cell pellets were resuspended in 200 µl DPBS + 0.1% BSA and kept on ice. Just prior to encapsulation, cells were passed through a 40 µm cell strainer, and cell counts and viability were measured. After droplet encapsulation, barcoding, and library preparation using the 10X Genomics Chromium Single Cell 3′ kit (version 3), data were sequenced on an Illumina NovaSeq 6000 sequencer.

FASTQ files were demultiplexed and aligned to the GRCz11 build of the zebrafish genome using Cellranger (version 6.1.0) (*Zheng et al., 2017*). Raw Cellranger count matrices were imported into R (version 4.1.2) using Seurat (version 4.1.0) (*Hao et al., 2021*). First, we reviewed data for quality and excluded any droplet that did not meet all of the following criteria: (1) have at least 1500 unique molecular identifiers (UMIs); (2) covering at least 750 distinct genes; (3) have <5% of genes mapping to the mitochondrial genome; and (4) have a log10 of detected genes per UMI >80%. After quality control, the dataset was also filtered to exclude genes with a detection rate below 1 in 3000 cells, leaving a total of 20,078 distinct genes expressed across 19,457 cells for analysis.

The quality-controlled count data were normalized using Pearson's residuals from the regularized negative binomial regression model, as implemented in Seurat::SCTransfrom (*Hafemeister and Satija, 2019*). When computing the SCT model, the effect of the total number of UMIs and number of detected genes per cell were regressed out. After normalization, the top 3000 most variably expressed genes were used to calculate principal components. Data were then integrated by source sample using Harmony (version 0.1.0) (*Korsunsky et al., 2019*). A two-dimensional uniform manifold approximation and projection (*Becht et al., 2018*) was then derived from the first 40 Harmony embeddings for visualization.

Using the integrated Harmony embeddings, clusters were defined with the Louvain clustering method, as implemented within Seurat. A resolution of 0.3 was used for cluster definition. Cluster identities were assigned by manually reviewing the results of Seurat::FindAllMarkers, searching for genes associated to known developmental lineages. Gene expression data for key markers that guided cluster identity assignment were visualized using Seurat::DotPlot.

Following this detailed annotation, some clusters were grouped to focus downstream analyses on three major lineages: ventral mesoderm (grouping cells from the pronephros, vasculogenic/myeloid precursors, hematopoietic cells, heart primordium, and cephalic mesoderm clusters), dorsal mesoderm (adaxial cells, segmental plate, and paraxial mesoderm), and ectoderm (CNS, mid/hindbrain boundary, spinal cord, and neural crest). For those three lineages, single-cell level data were aggregated per sample and cluster to perform pseudobulk DEA contrasting genotypes. Independent pairwise comparisons of *dact1−/−;dact2−/−* versus wildtype and *gpc4−/−* versus wildtype were performed using DESeq2 (v1.34.0) (*Love et al., 2014*). p-values were corrected for multiple testing using the default Benjamini–Hochberg method; log2 fold change values were corrected using the apeglm shrinkage estimator (*Zhu et al., 2019*). Significance was defined as an adjusted p-value <0.1 and log2 fold change >0.58 in absolute value. Heatmaps of the top most significant DEGs were generated from the regularized log transformed data using pheatmap (version 1.0.12). Overlap in significant genes across pairwise comparisons were determined and visualized in Venn diagrams. Over-representation analyses against the Gene Ontology database were ran using clusterProfiler (version 4.2.2) (*Wu et al., 2021*), using as input the set of genes found to be differentially expressed in the comparison of *dact1−/−;dact−/−* versus wildtype but not *gpc4−/−* versus wildtype.

## Statistical analysis

All sample sizes represent biological replicates. Analyses were performed using Prism Software (GraphPad) unless otherwise specified. An unpaired Student's *t* test or one-way ANOVA with multiple comparisons was used as indicated and a p-value <0.05 was considered significant. Graphs represent the mean ± SEM and *n* represents biological replicates. For categorical data (normal vs. mutant phenotype) a Fisher exact test was performed between *gfp* and *capn8* injected embryos and the odds ratio was determined. The confidence interval was determined by the Baptista–Pike method.

## Acknowledgements

We thank Christoph Seiler, Adele Donohue, and the Aqautics Facility team at Children's Hospital of Philadelphia; and Jessica Bethoney at Massachusetts General Hospital (MGH) for their excellent management of our zebrafish colonies and facilities. We thank the MGH Next Generation Sequencing Core for cell encapsulation, cDNA library preparation, and sequencing. Single-cell sequencing analysis

was performed by the Harvard Chan Bioinformatics Core. Work by AJ was funded in part by the Harvard Stem Cell Institute. We appreciate and acknowledge the generous funding support from the Shriners Hospitals for Children. This work was supported by the National Institutes of Health grant R01DE027983 to ECL.

## Additional information

### Funding

| Funder | Grant reference number | Author |
|---|---|---|
| National Institutes of Health | R01DE027983 | Eric C Liao |
| Shriners Hospitals for Children | | Eric C Liao |
| Harvard Stem Cell Institute | | Amelie M Jule |

The funders had no role in study design, data collection, and interpretation, or the decision to submit the work for publication.

### Author contributions

Shannon H Carroll, Conceptualization, Data curation, Investigation, Writing - original draft, Writing - review and editing; Sogand Schafer, Data curation, Writing - review and editing; Kenta Kawasaki, Casey Tsimbal, Shawn A Hallett, Edward Li, Data curation; Amelie M Jule, Formal analysis, Visualization, Methodology; Eric C Liao, Conceptualization, Supervision, Funding acquisition, Investigation, Writing - original draft, Project administration, Writing - review and editing

### Author ORCIDs

Shannon H Carroll http://orcid.org/0009-0008-8577-9012
Shawn A Hallett https://orcid.org/0000-0003-1472-7502
Eric C Liao https://orcid.org/0000-0001-6385-7448

### Ethics

All animal husbandry and experiments were performed in accordance with and approval from the Massachusetts General Hospital Institutional Animal Care and Use Committee (protocol number 2010N000106) and the Children's Hospital of Philadelphia Animal Care and Use Committee (protocol number IAC22001475). Zebrafish (Danio rerio) embryos and adults were maintained in accordance with institutional protocols.

Reviewer #2 (Public review): https://doi.org/10.7554/eLife.91648.4.sa1
Reviewer #3 (Public review): https://doi.org/10.7554/eLife.91648.4.sa2
Author response https://doi.org/10.7554/eLife.91648.4.sa3

## Additional files

### Supplementary files
• MDAR checklist

### Data availability

Sequencing data have been deposited in GEO under accession code GSE240264.

The following dataset was generated:

| Author(s) | Year | Dataset title | Dataset URL | Database and Identifier |
|-----------|------|---------------|-------------|-------------------------|
| Carroll SH, Schafer S, Kawasaki K, Tsimbal C, Julé AM, Hallett S, Li E, Liao EC | 2024 | dact1/2 modifies noncanonical Wnt signaling and calpain 8 expression to regulate convergent extension and craniofacial development | https://www.ncbi.nlm.nih.gov/geo/query/acc.cgi?acc=GSE240264 | NCBI Gene Expression Omnibus, GSE240264 |

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
