## [Editor Report · eLife Assessment]

This in several parts **valuable** study confirms the roles of Dact1 and Dact2, two factors involved in Wnt signaling, during zebrafish gastrulation and demonstrates their genetic interactions with other Wnt components to modulate craniofacial morphologies. Unfortunately, there are several limitations associated with the study, making it challenging to distinguish the primary and secondary effects of each factor, and their roles in craniofacial morphogenesis. The findings of a new potential target of dact1/2-mediated Wnt signaling are potentially of value; however, experimental evidence supporting their functional significance remains **incomplete** due to inconsistent results and limitations inherent to the overexpression approach.

---

## [Referee Report · Reviewer #2 (Public review)]

Summary:

Non-canonical Wnt signaling plays an important role in morphogenesis, but how different components of the pathway are required to regulate different developmental events remains an open question. This paper focuses on elucidating the overlapping and distinct functions of dact1 and dact2, two Dishevelled-binding scaffold proteins, during zebrafish axis elongation and craniofacial development. By combining genetic studies, detailed phenotypic analysis, lineage tracing, and single cell RNA-sequencing, the authors aimed to understand (1) the relative function of dact1/2 in promoting axis elongation, (2) their ability to modulate phenotypes caused by mutations in other non-canonical wnt components, and (3) pathways downstream of dact1/2.

Corroborating previous findings, this paper showed that dact1/2 is required for convergent extension during gastrulation and body axis elongation. Qualitative evidence was also provided to support dact1/2's role in genetically modulating non-canonical wnt signaling to regulate body axis elongation and the morphology of the ethmoid plate (EP). However, the spatiotemporal function of dact1/2 remains unknown. The use of scRNA-seq identified novel pathways and targets downstream of dact1/2. Calpain 8 is one such example, and its overexpression in some of the dact1/2+/- embryos was able to phenocopy the dact1/2−/− mutant EP morphology, pointing to its sufficiency in driving the EP phenotype in a few embryos. However, the same effect was not observed in dact1−/−; dact2+/- embryos, leading to the question of how significant calpain 8 really is in this context. The requirement of calpain 8 in mediating the phenotype is unclear as well. This is the most novel aspect of the paper, but some weaknesses remain in convincingly demonstrating the importance of calpain 8.

Strengths:

(1) The generation of dact1/2 germline mutants and the use of genetic approaches to dissect their genetic interactions with wnt11f2 and gpc4 provide unambiguous and consistent results that inform the relative functions of dact1 and dact2, as well as their combined effects.

(2) Because the ethmoid plate exhibits a spectrum of phenotypes in different wnt genetic mutants, it is a useful system for studying how tissue morphology can be modulated by different components of the wnt pathway.

(3) The authors leveraged lineage tracing by photoconversion to dissect how dact1/2 differentially impacts the ability of different cranial neural crest populations to contribute to the ethmoid plate. This revealed that distinct mechanisms via dact1/2 and shh can lead to similar phenotypes.

(4) The use of scRNA-seq was a powerful approach and identified potential novel pathways and targets downstream of dact1/2.

Weaknesses:

(1) Connecting the expression of dact1/2 and wnt11f2 to their mutant phenotypes: Given that dact1/2 and wnt11f2 expression are quite distinct, at least in the stages examined, the claim that dact1/2 function downstream of wnt11f2 is not well supported. That conclusion was based on shared craniofacial phenotypes between dact1/2−/−, wnt11f2−/−, and dact1/2−/−;wnt11f2−/− mutants. However, because the craniofacial phenotype is likely a secondary effect of dact1/2 deletion, using it to interpret the signaling axis between dact1/2 and wnt11f2 is not appropriate.

(2) Spatiotemporal function of dact1/2: Germline mutations limit the authors' ability to study a gene's spatiotemporal functional requirement. They, therefore, cannot concretely attribute nor separate early-stage phenotypes (during gastrulation) to/from late stage phenotypes (EP morphological changes), which the authors postulated to result from secondary defects in floor plate and eye field morphometry. As a result, whether dact1/2 are directly involved in craniofacial development is not addressed, and the mechanisms resulting in the craniofacial phenotypes are also unclear.

(3) The functional significance of calpain 8: Because calpain 8 was upregulated in many dact1/2−/− mutant cell populations (although not in the neural crest) during gastrulation, the authors tested its function by overexpressing capn8 mRNA in embryos. While only 1 out of 142 calpain 8-overexpressing wild type animals phenocopied dact1/2 mutants, 7.5% of dact1/2+/- embryos overexpressing capn8 exhibited dact1/2-like phenotypes. However, the same effect was not observed in dact1−/−; dact2+/- embryos. Given the expression pattern of calpain 8 and results from the overexpression study, the function of capn8 remains inconclusive. The requirement of calpain 8 in driving the phenotype remains unclear. The authors stated these limitations in their study.

---

## [Referee Report · Reviewer #3 (Public review)]

Summary:

In this manuscript the authors explore the roles of dact1 and dact2 during zebrafish gastrulation and craniofacial development. Previous studies used morpholino (MO) knockdowns to show that these scaffolding proteins, which interact with dissheveled (Dsh), are expressed during zebrafish gastrulation and suggested that dact1 promotes canonical Wnt/B-catenin signaling, while dact2 promotes non-canonical Wnt/PCP-dependent convergent-extension (Waxman et al 2004). This study goes beyond this work by creating loss-of-function mutant alleles for each gene and unlike the MO studies finds little (dact2) to no (dact1) phenotypic defects in the homozygous mutants. Interestingly, dact1/2 double mutants have a more severe phenotype, which resembles those reported with MOs as well as homozygous wnt11/silberblick (wnt11/slb) mutants that disrupt non-canonical Wnt signaling (Heisenberg et al., 1997; 2000). Further analyses in this paper try to connect gastrulation and craniofacial defects in dact1/2 mutants with wnt11/slb and other wnt-pathway mutants. scRNAseq conducted in mutants identifies calpain 8 as a potential new target of dact1/2 and Wnt signaling.

Previous comments:

Strengths:

When considered separately the new mutants are an improvement over the MOs and the paper contains a lot of new data.

Weaknesses:

However, the hypotheses are very poorly defined and misinterpret key previous findings surrounding the roles of wnt11 and gpc4, which results in a very confusing manuscript. Many of the results are not novel and focus on secondary defects. The most novel result overexpressing calpain8 in dact1/2 mutants is preliminary and not convincing.

The authors addressed some of our comments, but not our main criticisms, which we reiterate here:

(1) The authors argue that morpholino studies are unreliable and here they made new mutants to solve this uncertainty for dap 1/2. However, creating stable mutant lines to largely confirm previous results obtained by using morpholino knock-down phenotypes does not justify publication in eLife.

(2) The authors argue that since it has not been shown conclusively that craniofacial defects in wnt11 and dap1/2 mutants are secondary to gastrulation defects there is no solid evidence preventing them from investigating these craniofacial defects. However, since it is extremely likely that the rod-like ethmoid plates of wnt11f2- and dact1/2 mutants focused on here are secondary to gastrulation defects previously described by others (Heisenberg and NussleinVolhard 1997; Waxman et al., 2004), the burden of proof is on the authors to provide much stronger evidence against this interpretation.

(3) The data for calpain overexpression remains too preliminary.

---

## [Author Response]

The following is the authors’ response to the previous reviews.

**Reviewer #1 (Recommendations For The Authors):**
This is not a recommendation. While reading old literature, I found some interesting facts. The shape of the neurocranium in monotremes, birds, and mammals, at least in early stages, resembles the phenotype of 'dact'1/2, wnt11f2, or syu mutants. For more details, see DeBeer's: 'The Development of the Vertebrate Skull, !937' Plate 137.

Thank you for pointing this out. It is indeed interesting.

Minor Comments:• Lines 64, 66, and 69: same citation without interruption: Heisenberg, Brand et al. 1996

Revised line 76.

• Lines 101 and 102: same citation without interruption: Li, Florez et al. 2013

Revised line 118.

• Lines 144, 515, 527, and 1147: should be wnt11f2 instead of wntllf2 - if not, then explain

Revised lines 185, 625, 640,1300.

• Lines 169 and 171: incorrect figure citation: Fig 1D - correct to Fig 1F

Revised lines 217, 219.

• Line 173: delete (Fig. S1)

Revised line 221.

• Line 207: indicate that both dact1 and dact2 mRNA levels increased, noting a 40% higher level of dact2 mRNA after deletion of 7 bp in the dact2 gene

Revised line 265.

• Line 215: Fig 1F instead of Fig 1D

Revised line 217.

• Line 248: unify naming of compound mutants to either dact1/2 or dact1/dact2 compound mutants

Revised to dact1/2 throughout.

• Line 259: incorrect figure citation: Fig S1 - correct to Fig S2D/E

Revised line 324.

• Line 302: correct abbreviation position: neural crest (NCC) cell - change to neural crest cell (NCC) population

Revised line 380.

• Line 349: repeating kny mut definition from line 70 may be unnecessary

Revised line 434.

• Line 351: clarify distinction between Fig S1 and Fig S2 in the supplementary section

Revised line 324.

• Line 436: refer to the correct figure for pathways associated with proteolysis (Fig 7B)

Revised line 530.

• Line 446-447: complete the sentence and clarify the relevance of smad1 expression, and correct the use of "also" in relation to capn8

Revised line 567.

• Line 462: clarify that this phenotype was never observed in wildtype larvae, and correct figure reference to exclude dact1+/- dact2+/-

Revised line 563, 568.

• Line 463: explain the injection procedure into embryos from dact1/2+/- interbreeding

Revised line 565.

• Lines 488 and 491: same citation without interruption: Waxman, Hocking et al. 2004

Revised line 591.

• Line 502: maintain consistency in referring to TGF-beta signaling throughout the article

Revised throughout.

• Line 523: define CNCC; previously used only NCC

Revised to cranial NCC throughout.

• Line 1105: reconsider citing another work in the figure legend

Revised line 1249.

• Line 1143: consider using "mutant" instead of "mu"

Revised line 1295.

• Fig 2A/B: indicate the number of animals used ("n")

N is noted on line 1274.

• Fig 2C, D, E: ensure uniform terminology for control groups ("wt" vs. "wildtype")

Revised in figure.

• Fig 7C: clarify analysis of dact1/2−/− mutant in lateral plate mesoderm vs. ectoderm

Revised line 1356.

• Fig 8A: label the figure to indicate it shows capn8, not just in the legend

Revised.

• Fig 8D: explain the black/white portions and simplify to highlight important data

Revised.

• Fig S2: add the title "Figure S2"

Revised.

• Consider omitting the sentence: "As with most studies, this work has contributed some new knowledge but generated more questions than answers."

Revised line 720.

**Reviewer #2 (Recommendations For The Authors):**
Major comments:(1) The authors have addressed many of the questions I had, including making the biological sample numbers more transparent. It might be more informative to use n = n/n, e.g. n = 3/3, rather than just n = 3. Alternatively, that information can be given in the figure legend or in the form of penetrance %.

The compound heterozygote breeding and phenotyping analyses were not carried out in such a way that we can comment on the precise % penetrance of the ANC phenotype, as we did not dissect every ANC and genotype every individual that resulted from the triple heterozygote in crossings. We collected phenotype/genotype data until we obtained at least three replicates.

We did genotype every individual resulting from dact1/2 dHet crosses to correlate genotype to the phenotype of the embryonic convergent extension phenotype and narrowed ethmoid plate (Fig. 2A, Fig. 3) which demonstrated full penetrance.

(2) The description of the expression of dact1/2 and wnt11f2 is not consistent with what the images are showing. In the revised figure 1 legend, the author says "dact2 and wnt11f2 transcripts are detected in the anterior neural plate" (line 1099)", but it's hard to see wnt11f2 expression in the anterior neural plate in 1B. The authors then again said " wnt11f2 is also expressed in these cells", referring to the anterior neural plate and polster (P), notochord (N), paraxial and presomitic mesoderm (PM) and tailbud (TB). However, other than the notochord expression, other expression is actually quite dissimilar between dact2 and wnt11f2 in 1C. The authors should describe their expression more accurately and take that into account when considering their function in the same pathway.

We have revised these sections to more carefully describe the expression patterns. We have added references to previous descriptions of wnt11 expression domains.

(3) Similar to (2), while the Daniocell was useful in demonstrating that expression of dact1 and dact2 are more similar to expression of gpc4 and wnt11f2, the text description of the data is quite confusing. The authors stated "dact2 was more highly expressed in anterior structures including cephalic mesoderm and neural ectoderm while dact1 was more highly expressed in mesenchyme and muscle" (lines 174-176). However, the Daniocell seems to show more dact1 expression in the neural tissues than dact2, which would contradict the in situ data as well. I think the problem is in part due to the dataset contains cells from many different stages and it might be helpful to include a plot of the cells at different stages, as well as the cell types, both of which are available from the Daniocell website.

We have revised the text to focus the Daniocell analysis on the overall and general expression patterns. Line 220.

(4) The authors used the term "morphological movements" (line 337) to describe the cause of dact1/2 phenotypes. Please clarify what this means. Is it cell movement? Or is it the shape of the tissues? What does "morphological movements" really mean and how does that affect the formation of the EP by the second stream of NCCs?

We have revised this sentence to improve clarity. Line 416.

(5) In the first submission, only 1 out of 142 calpain-overexpressing animals phenocopied dact1/2 mutants and that was a major concern regarding the functional significance of calpain 8 in this context. In the revised manuscript, the authors demonstrated that more embryos developed the phenotype when they are heterozygous for both dact1/2. While this is encouraging, it is interesting that the same phenomenon was not observed in the dact1−/−; dact2+/- embryos (Fig. 6D). The authors did not discuss this and should provide some explanation. The authors should also discuss sufficiency vs requirement tested in this experiment. However, given that this is the most novel aspect of the paper, performing experiments to demonstrate requirements would be important.

We have added a statement regarding the non-effect in dact1−/−;dact2+/- embryos. Line 568-570. We have also added discussion of sufficiency vs necessity/requirement testing. Line 676-679.

(6) Related to (5), the authors cited figure 8c when mentioning 0/192 gfp-injected embryos developed EP phenotypes. However, figure 8c is dact1/2 +/- embryos. The numbers also doesn't match the numbers in Figure 8d either. Please add relevant/correct figures.

The text has been revised to distinguish between our overexpression experiment in wildtype embryos (data not shown) versus overexpression in dact1/2 double het in cross embryos (Fig 8).

Minor comments:(1) Fig 1 legend line 1106 "the midbrain (MP)" should be MB

Revised line 1250.

(2) Wntllf2, instead of wnt11f2, (i.e. the letter "l" rather than the number "1") was used in 4 instances, line 144, 515, 527, 1147

Revised lines 185, 625, 640,1300.

(3) The authors replaced ANC with EP in many instances, but ANC is left unchanged in some places and it's not defined in the text. It's first mentioned in line 170.

Revised line 218.